# Conversational Drug Editing Using Retrieval and Domain Feedback

**Shengchao Liu**[1] *,   **Jiongxiao Wang**[2] *,   **Yijin Yang**[3],   **Chengpeng Wang**[4],   **Ling Liu**[5],
**Hongyu Guo**[6,7] †,   **Chaowei Xiao**[2] †

[1]University of California, Berkeley    [2]University of Wisconsin-Madison    [3]Arizona State University
[4]University of Illinois Urbana-Champaign    [5]Princeton University
[6]National Research Council Canada    [7]University of Ottawa
* Equal Contribution        † Joint Advising

## ABSTRACT

Recent advancements in conversational large language models (LLMs), such as ChatGPT, have demonstrated remarkable promise in various domains, including drug discovery. However, existing works mainly focus on investigating the capabilities of conversational LLMs on chemical reactions and retrosynthesis. While drug editing, a critical task in the drug discovery pipeline, remains largely unexplored. To bridge this gap, we propose ChatDrug, a framework to facilitate the systematic investigation of drug editing using LLMs. ChatDrug jointly leverages a prompt module, a retrieval and domain feedback module, and a conversation module to streamline effective drug editing. We empirically show that ChatDrug reaches the best performance on all 39 drug editing tasks, encompassing small molecules, peptides, and proteins. We further demonstrate, through 10 case studies, that ChatDrug can successfully identify the key substructures for manipulation, generating diverse and valid suggestions for drug editing. Promisingly, we also show that ChatDrug can offer insightful explanations from a domain-specific perspective, enhancing interpretability and enabling informed decision-making.

## 1 INTRODUCTION

Recently, artificial intelligence (AI) tools have made remarkable strides in revolutionizing the field of drug discovery, offering tremendous potential for accelerating and enhancing various stages of the process (Sullivan, 2019), including but not limited to virtual screening (Rohrer & Baumann, 2009; Liu et al., 2018), lead optimization (Jin et al., 2020; Irwin et al., 2022; Wang et al., 2022; Liu et al., 2022b), reaction and retrosynthesis (Gottipati et al., 2020), and protein folding and inverse folding (Jumper et al., 2021; Hsu et al., 2022). However, existing research has predominantly focused on the drug structure information, solely considering the inherent chemical structure of the drugs as a single modality. In contrast, the drug discovery pipeline involves iterative refining processes that entail conversations with domain experts to incorporate their feedback, ultimately achieving the desired outcome. On the other hand, significant advancements have been made in large language models (LLMs) (Brown et al., 2020; Devlin et al., 2018; Yang et al., 2019), showcasing exceptional capabilities in understanding human knowledge and exhibiting promising reasoning abilities (Huang et al., 2022; Zhou et al., 2022; Kojima et al., 2022). Such observations inspire us to investigate the potential of leveraging LLMs' conversation and reasoning abilities for AI-assisted drug discovery in a multi-modality fashion.

**Potential of Conversational LLMs for Drug Discovery and Editing.** Conversational LLMs exhibit three compelling factors that make them highly promising for drug discovery. Firstly, these model  (Taylor et al., 2022; Touvron et al., 2023) are pretrained on a comprehensive knowledge base, enabling their application across various fields, including drug discovery. This extensive "world-level" knowledge serves as a robust foundation for drug-related tasks. Second, conversational LLMs possess outstanding abilities in fast adaptation and generalization. For example, by leveraging few-shot demonstrations, these models can effectively activate the relevant concepts learned during pretraining, enabling them to deliver more accurate and desired answers (Xie et al., 2021). Such adaptability and generalization capacity holds immense potential for addressing complex drug discovery challenges and generating valuable insights. Lastly, interactive communication is a vital characteristic of

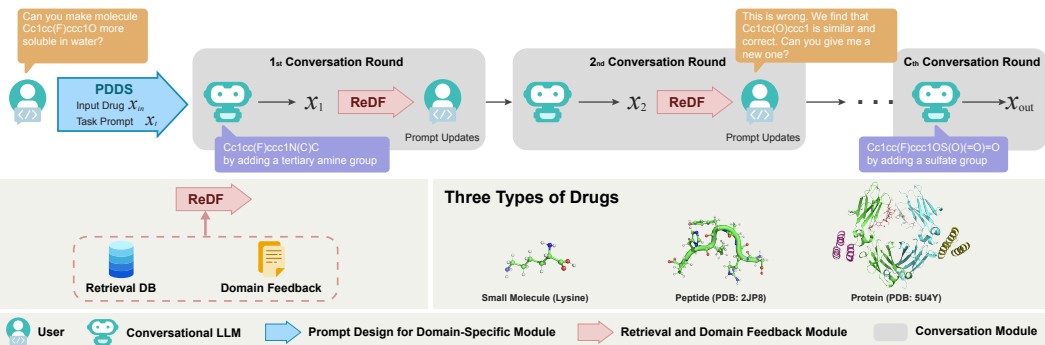

Figure 1: The pipeline for ChatDrug with 3 modules. PDDS generates drug editing prompts. ReDF updates the prompts using retrieved information and domain feedback. Finally, ChatDrug adopts the conversational module for interactive refinement. Further, we demonstrate 3 drug types: small molecules, peptides, and proteins.

conversational LLMs. This feature allows for a dynamic exchange of information, enabling users to incorporate feedback from prior knowledge or domain experts into the model. This bidirectional flow of information facilitates self-calibration of the answers, leading to improved accuracy and relevance in drug discovery tasks. To sum up, these factors collectively highlight the untapped potential of conversational LLMs for drug discovery tasks. Noticeably, **drug editing** (AKA *lead optimization* or *protein design*) is an important and challenging task in drug discovery. This is a routine task in pharmaceutical companies, and it aims at updating the drug's substructures (Mihalić & Trinajstić, 1992), related to certain key tactics in drug discovery like functional group change (Ertl et al., 2020) and scaffold hopping (Böhm et al., 2004; Hu et al., 2017). Traditional solutions relying on domain experts for manual editing can be subjective or biased (Drews, 2000; Gomez, 2018). Recent works (Liu et al., 2023b; 2022a) have started to explore text-guided drug editing in a multi-modal manner. However, they do not utilize the conversational potentials in LLMs (Peng et al., 2023).

**Our Approach: ChatDrug.** Motivated by the aforementioned factors and challenges, we propose ChatDrug, a framework aiming to unlock new possibilities and enhance drug editing by using LLMs. Such a framework can be applied to various LLMs backbones, including ChatGPT and open-source LLMs model GALACTICA and Llama2, and the pipeline is shown in Figure 1. In ChatDrug, users can activate the conversation with LLMs involving domain knowledge and inject the retrieved information into the conversation. Specifically, ChatDrug includes the following modules for conversational drug editing. First, ChatDrug adopts a PDDS (prompt design for domain-specific) module, enabling strong prompt engineering capability from LLMs. Second, ChatDrug integrates a ReDF (retrieval and domain feedback) module. By leveraging the vast domain knowledge available, such a ReDF module serves as guidance for prompt updates and augments the model's performance in generating accurate outputs. Third, ChatDrug adopts a conversation-based approach, aligning with the iterative refinement nature of the drug discovery pipeline. Such interactive schema enables a dynamic and collaborative process, effectively incorporating feedback from domain experts to achieve desired outcomes.

Through our design, ChatDrug demonstrates two appealing properties for drug editing tasks: (1) ChatDrug exhibits an open vocabulary property, allowing for exploring novel drug concepts beyond a fixed set of pre-defined annotations. The model can generalize to new drug-related concepts due to the unrestricted nature of natural language. (2) ChatDrug possesses the compositional property. It can decompose complex concepts, such as multi-objective lead optimization, into simpler attributes like binding to a new protein and high permeability, enabling handling complicated drug editing tasks. [1]

Then, to fully verify the effectiveness of ChatDrug, we need to design a benchmark for a wide range of drug editing tasks. Before going into details, we would like to claim two criteria for the task design: (1) The tasks should involve indeterministic answers, as they can serve as a source of inspiration for domain experts. (2) The tasks should be able to evaluate computationally since the lab experiment can be quite laborious and is beyond the discussion of this paper. Following these criteria, we introduce 39 editing tasks over three common drugs: 28 for small molecules, 9 for peptides, and 2 for proteins.

---

[1]Note that ChatDrug aims to inspire domain experts rather than replace them. While ChatDrug can propose optimized drugs or novel attributes, its primary role is to serve as a tool for knowledge exploration. The generated outputs can provide valuable insights and spark inspiration for domain experts in the drug discovery process.

Last but not least, we offer empirical evidence substantiating the capability of ChatDrug for a wide range of drug editing tasks on three LLM backbones: Turbo (a.k.a. ChatGPT), GALACTICA and Llama2. Quantitatively, ChatDrug reaches the best performance on all 39 drug editing tasks compared to seven baselines, among which, ChatDrug-Turbo reaches generally better performance and higher stability on 32 of them. Thus, we further qualitatively provide 10 case studies illustrating that ChatDrug-Turbo can successfully identify the important substructures for each type of drug, as follows. (1) For small molecules, ChatDrug-Turbo detects the key scaffold for molecule editing, such as changing polar or hydrophobic functional groups for tuning properties like solubility in water and permeability. (2) For peptides, ChatDrug-Turbo accurately identifies the protein-specific binding motifs of the peptide sequences. (3) For proteins, ChatDrug-Turbo modifies sequences with more $\alpha$-helix or $\beta$-strand structures after folding (Jumper et al., 2021; Lin et al., 2022). We additionally illustrate that ChatDrug provides insightful explanations, serving as a knowledge extraction tool.

## 2 PRELIMINARIES

**Data Structure of Drugs.** Drugs (Wishart et al., 2008; dru) refer to certain specific substances that can be adopted to prevent, diagnose, treat, or relieve symptoms of a disease or abnormal condition. In this paper, we would like to explore the three most common drugs: small molecules (Jayatunga et al., 2022), proteins (Frokjaer & Otzen, 2005), and peptides (Craik et al., 2013). Small molecules are sets of atoms connected together through the covalent bonds. Commonly-used data structures include SMILES (simplified molecular-input line-entry system) strings (Weininger, 1988) and molecular graphs (Duvenaud et al., 2015; Kearnes et al., 2016; Liu et al., 2019). In ChatDrug, we consider using the SMILES strings. Proteins are complex macromolecules, and they are composed of 20 amino acids, where each amino acid is a small molecule. Regarding the protein data structure, we adopt the amino acid sequence (*i.e.*, amino acid string), and the mapping between 20 alphabets and amino acids can be found in Appendix E. Peptides are short chains of amino acids and can be viewed as a special type of protein. The demonstration of three data structures can be found in Figure 1.

**Drug Editing and Problem Formulation.** In this paper, we focus on the drug editing task. Drug editing is also known as *lead optimization* or *protein design*, an important drug discovery task. From the machine learning perspective, drug editing is a **conditional generation** problem and can be formulated as follows. Suppose the input drug (SMILES string or amino acid sequence) is $x_{\text{in}}$, and a target or desired property in the textual description is also known as the **text prompt** $x_t$ in literature (Raffel et al., 2020; Liu et al., 2023a). Then condition on such text prompt, the goal is to obtain an optimized drug $x_{\text{out}} \sim P(x|x_{\text{in}}, x_t)$. Under the ChatDrug framework, the formulation is:

$$x_{\text{out}} = \text{ChatDrug}(x_{\text{in}}, x_t). \tag{1}$$

Then an evaluation metric $E(x_{\text{in}}, x_{\text{out}}; x_t) \in \{\text{True}, \text{False}\}$ is applied to check if the edited drugs can satisfy the desired properties compared to the input drugs, and we will average this over each corresponding task to get the *hit ratio*. Note that $E(\cdot, \cdot; \cdot)$ is task-specific, as will be discussed in Section 4.

## 3 METHOD: THE CHATDRUG FRAMEWORK

**Overview.** Our ChatDrug framework is illustrated in Figure 1. It consists of three components: (1) Prompt Design for Domain-Specific (PDDS) module, (2) Retrieval and Domain Feedback (ReDF) module, and (3) conversation module. Given a task prompt and input drug, PDDS aims to generate the domain-specific prompt and concatenate it with the input drug to request LLMs for answers. One problem for current LLMs is that it does not fully utilize the prior domain knowledge. Thus, we design the ReDF module aiming to (1) guide the LLMs to solve this task by retrieving structurally similar examples from the database and adding examples into the prompt as demonstrations and (2) verify the correctness of the output by using a domain feedback function. If the output drug is incorrect after ReDF, we then adopt the conversation module to ask LLMs to generate a new drug iteratively. We highlight that ChatDrug is a parameter-free scheme and *does not require any learning* procedure.

### 3.1 PDDS MODULE

ChatDrug is proposed to solve a challenging problem: generalization of a universally (w.r.t. data type and data source) well-trained LLM to solving scientific tasks. In natural language processing (NLP), prompt design or prompt engineering (Liu et al., 2023a) has proven to be an effective paradigm for

generalizing well-trained LLMs to various NLP downstream tasks, including but not limited to sentiment classification (Han et al., 2022; Hu et al., 2021), textual entailment (Webson & Pavlick, 2021; Shin et al., 2020), text summarization (He et al., 2020; Schick & Schütze, 2020; Dou et al., 2020).

However, the explorations of adapting LLMs for drug editing tasks have been lagging behind. In this paper, we investigate this problem in the three most common types of drugs: small molecules, protein-binding peptides, and proteins. Recall that the goal of ChatDrug is (as in Equation (1)): $x_{\text{out}} = \text{ChatDrug}(x_{\text{in}}, x_t)$. Here, the text prompts $x_t$ are specifically designed to enable the generalization for domain-specific tasks with computationally feasible metrics. Additionally, we want to highlight that the objectives for drug editing (in $x_t$) should be related to the *high-level property* instead of *exact substructure replacement* for two main reasons. (1) As discussed in Appendix C, ChatDrug suits better for fuzzy matching like edited drugs with desired properties. In contrast, exact substructure replacement can be easily and precisely performed by domain experts, and such replacement may lack the creative inspiration for humans. (2) Property-related questions have an ambiguous nature, leading to dispersed answers that spark inspiration for domain experts in the drug discovery process.

Then concretely on the prompt design, for small molecules, we consider properties like solubility, drug-likeness, permeability, and the number of acceptors/donors. For peptides, we consider the properties of peptide-MHC binding. For proteins, we consider the secondary structure. The text prompts are to explicitly depict the desired properties to be either higher or lower, and corresponding task prompts will be briefly explained in Section 4. One concrete example for molecule editing is *"Can you make molecule [$x_{in}$] more soluble in water."*, and more details can be found in Appendix F.

## 3.2 REDF MODULE

To better utilize the domain knowledge, we propose an important module: the ReDF (retrieval and domain feedback) module. The intuition is that there exists rich domain knowledge in the form of a retrieval database (DB), and ReDF will retrieve the useful information and inject it into the text prompt, adopting the fascinating language understanding ability of conversational LLMs.

Specifically, for each input drug $x_{\text{in}}$ and prompt $x_t$, we have a candidate drug $\tilde{x}$, which does not satisfy the desired property change in $x_t$. The candidate drug has diverse data resources based on the problem setup, and in ChatDrug, it is the output drug with the negative result at each conversation round (will be introduced in Section 3.3). Then, the ReDF module returns a drug $x_R$ satisfying:

$$x_R = \text{ReDF}(x_{\text{in}}, \tilde{x}; x_t) = \underset{x'_R \in \text{Retrieval DB}}{\arg\max} \langle \tilde{x}, x'_R \rangle \wedge D(x_{\text{in}}, x'_R; x_t), \qquad (2)$$

where $D(\cdot, \cdot; \cdot) \in \{\text{True}, \text{False}\}$ is the domain feedback function, and $\langle \tilde{x}, x'_R \rangle$ is the similarity function. We use Tanimoto similarity (Bajusz et al., 2015) for small molecules and Levenshtein distance for peptides and proteins. Notice that here we take $D(\cdot, \cdot; \cdot)$ the same as evaluation metric $E(\cdot, \cdot; \cdot)$, while there is some critical difference on the task-specific thresholds, as will be discussed in the ablation study in Section 4.5. Then the ReDF module injects $x_R$ into a new prompt, *e.g.*, the updated prompt for a molecule task is *"Your provided sequence [$\tilde{x}$] is not correct. We find a sequence [$x_R$] which is correct and similar to the molecule you provided. Can you give me a new molecule?"*

We also want to highlight that the domain feedback injection in ReDF is relevant to the *in-context learning (ICL)* (Dong et al., 2022). Such knowledge injection can result in performance gain (Min et al., 2022) not only because of the mapping between ground truth data-label pairs but also due to the demonstration of the in-distribution data and label space. An ablation study on this is in Section 4.

## 3.3 CONVERSATION MODULE

Another appealing attribute of conversational LLMs (like ChatGPT) is their interactive capability. This enables the LLMs to iteratively update the results by injecting prior knowledge. Inspired by this, we also consider adapting the conversational strategy for ChatDrug, which can naturally fit the ReDF module as described in in Section 3.2. Then, concretely on this conversational strategy in ChatDrug, first suppose there are $C$ conversation rounds, and we have an edited drug $x_c$ for the conversation round $c$. If $x_c$ satisfies our condition in the task prompt, then ChatDrug will exit. Otherwise, users will tell ChatDrug that $x_c$ is wrong, and we need to retrieve another similar and correct drug from the retrieval DB using ReDF: $x_R = \text{ReDF}(x_{\text{in}}, x_c)$, with $\tilde{x} = x_c$ in Equation (2).

Table 1: Results on 16 single-objective small molecule editing, and the evaluation is the hit ratio of the property change. For ChatDrug, we report the mean and std of five random seeds. The best results are marked in **bold**.

| Single Target Property | Δ | Random | PCA | High Variance | GS-Mutate | MoleculeSTM | | ChatDrug | | |
|---|---|---|---|---|---|---|---|---|---|---|
| | | | | | | SMILES | Graph | GALACTICA | Llama2 | Turbo |
| 101 *more soluble in water* | 0 | 35.33 ± 1.31 | 33.80 ± 3.63 | 33.52 ± 3.75 | 52.00 ± 0.41 | 61.87 ± 2.67 | 67.86 ± 3.46 | 83.32 ± 1.13 | 42.88 ± 1.83 | **94.13±1.04** |
| | 0.5 | 11.04 ± 2.40 | 10.66 ± 3.24 | 10.86 ± 2.56 | 14.67 ± 0.62 | 49.02 ± 1.84 | 54.44 ± 3.99 | 78.20 ± 1.85 | 30.31 ± 1.91 | **88.67±0.95** |
| 102 *less soluble in water* | 0 | 43.36 ± 3.06 | 39.36 ± 2.55 | 42.89 ± 2.36 | 47.50 ± 0.41 | 52.71 ± 1.67 | 64.79 ± 2.76 | 72.41 ± 4.44 | 47.89 ± 2.05 | **96.86±1.10** |
| | 0.5 | 19.75 ± 1.56 | 15.12 ± 2.93 | 18.22 ± 0.33 | 12.50 ± 0.82 | 30.47 ± 3.26 | 47.09 ± 3.42 | 61.47 ± 3.55 | 34.76 ± 2.46 | **70.08±3.44** |
| 103 *more like a drug* | 0 | 38.06 ± 2.57 | 33.99 ± 3.72 | 36.20 ± 4.34 | 28.00 ± 0.71 | 36.52 ± 2.46 | 39.97 ± 4.32 | 41.49 ± 2.66 | 32.88 ± 0.99 | **48.65±3.39** |
| | 0.1 | 5.27 ± 0.24 | 3.97 ± 0.10 | 4.44 ± 0.58 | 6.33 ± 2.09 | 8.81 ± 0.82 | 14.06 ± 3.18 | **21.23 ± 0.96** | 12.38 ± 3.56 | 19.37±5.54 |
| 104 *less like a drug* | 0 | 36.96 ± 2.25 | 35.17 ± 2.61 | 39.99 ± 0.57 | 71.33 ± 0.85 | 58.59 ± 1.01 | 77.62 ± 2.80 | **92.13 ± 1.29** | 54.96 ± 1.26 | 70.75±2.92 |
| | 0.1 | 6.16 ± 1.87 | 5.26 ± 0.95 | 7.56 ± 0.29 | 27.67 ± 3.79 | 37.56 ± 1.76 | 54.22 ± 3.12 | **85.83 ± 1.76** | 29.13 ± 2.17 | 30.99±2.66 |
| 105 *higher permeability* | 0 | 25.23 ± 2.13 | 21.36 ± 2.59 | 21.98 ± 3.77 | 22.00 ± 0.82 | 57.74 ± 0.60 | 59.84 ± 0.78 | **84.21 ± 4.07** | 28.47 ± 2.76 | 56.56±1.84 |
| | 10 | 17.41 ± 1.43 | 14.52 ± 0.80 | 14.66 ± 2.13 | 6.17 ± 0.62 | 47.51 ± 1.88 | 50.42 ± 2.73 | **79.91 ± 1.25** | 25.69 ± 1.20 | 43.08±2.95 |
| 106 *lower permeability* | 0 | 16.79 ± 2.54 | 15.48 ± 2.40 | 17.10 ± 1.14 | 28.83 ± 1.25 | 34.13 ± 0.59 | 31.76 ± 0.97 | 68.49 ± 2.34 | 47.58 ± 3.62 | **77.35±1.98** |
| | 10 | 11.02 ± 0.71 | 10.62 ± 1.86 | 12.01 ± 1.01 | 15.17 ± 1.03 | 26.48 ± 0.97 | 19.76 ± 1.31 | 60.46 ± 0.57 | 38.58 ± 4.49 | **66.69±2.74** |
| 107 *more hydrogen bond acceptors* | 0 | 12.64 ± 1.64 | 10.85 ± 2.29 | 11.78 ± 0.15 | 21.17 ± 3.09 | 54.01 ± 5.26 | 37.35 ± 0.79 | 55.18 ± 3.67 | 10.70 ± 0.82 | **95.35±0.62** |
| | 1 | 0.69 ± 0.01 | 0.90 ± 0.84 | 0.67 ± 0.01 | 1.83 ± 0.47 | 27.33 ± 2.62 | 16.13 ± 2.87 | 37.40 ± 1.84 | 7.18 ± 1.76 | **72.60±2.51** |
| 108 *more hydrogen bond donors* | 0 | 2.97 ± 0.61 | 3.97 ± 0.55 | 6.23 ± 0.66 | 19.50 ± 2.86 | 28.55 ± 0.76 | 60.97 ± 5.09 | 59.41 ± 4.07 | 12.77 ± 4.62 | **96.54±1.31** |
| | 1 | 0.00 ± 0.00 | 0.00 ± 0.00 | 0.00 ± 0.00 | 1.33 ± 0.24 | 7.69 ± 0.56 | 32.35 ± 2.57 | 31.88 ± 3.22 | 7.15 ± 2.81 | **76.43±3.32** |

Table 2: Results on 12 multi-objective small molecule editing, and the evaluation is the hit ratio of the property change. For ChatDrug, we report the mean and std of five random seeds. The best results are marked in **bold**.

| Two Target Property | Δ | Random | PCA | High Variance | GS-Mutate | MoleculeSTM | | ChatDrug | | |
|---|---|---|---|---|---|---|---|---|---|---|
| | | | | | | SMILES | Graph | GALACTICA | Llama2 | Turbo |
| 201 *more soluble in water* and *more hydrogen bond acceptors* | 0 – 0 | 9.88 ± 1.03 | 8.64 ± 2.06 | 9.09 ± 1.25 | 14.00 ± 2.48 | 27.87 ± 3.86 | 27.43 ± 3.41 | 39.51 ± 3.41 | 24.95 ± 2.55 | **79.62±0.64** |
| | 0.5 – 1 | 0.23 ± 0.33 | 0.45 ± 0.64 | 0.22 ± 0.31 | 0.67 ± 0.62 | 8.80 ± 0.04 | 11.10 ± 1.80 | 26.44 ± 1.07 | 13.24 ± 1.17 | **49.64±2.66** |
| 202 *less soluble in water* and *more hydrogen bond acceptors* | 0 – 0 | 2.99 ± 0.38 | 2.00 ± 0.58 | 2.45 ± 0.67 | 7.17 ± 0.85 | 8.55 ± 2.75 | 8.21 ± 0.81 | 28.40 ± 3.11 | 8.91 ± 1.06 | **51.59±3.79** |
| | 0.5 – 1 | 0.45 ± 0.32 | 0.00 ± 0.00 | 0.22 ± 0.31 | 0.17 ± 0.24 | 2.93 ± 0.30 | 0.00 ± 0.00 | 12.66 ± 1.40 | 8.27 ± 1.66 | **24.92±4.85** |
| 203 *more soluble in water* and *more hydrogen bond donors* | 0 – 0 | 2.28 ± 1.15 | 2.23 ± 1.16 | 4.44 ± 0.58 | 13.83 ± 2.95 | 33.51 ± 4.08 | 49.23 ± 1.71 | 47.91 ± 3.33 | 30.66 ± 2.39 | **89.34±0.96** |
| | 0.5 – 1 | 0.00 ± 0.00 | 0.00 ± 0.00 | 0.00 ± 0.00 | 0.00 ± 0.00 | 9.98 ± 1.03 | 23.94 ± 1.09 | 26.49 ± 3.37 | 8.17 ± 3.34 | **53.64±5.81** |
| 204 *less insoluble in water* and *more hydrogen bond donors* | 0 – 0 | 0.69 ± 0.58 | 1.96 ± 0.87 | 1.79 ± 0.66 | 5.67 ± 0.62 | 17.03 ± 2.75 | 14.42 ± 3.43 | 25.70 ± 2.07 | 16.30 ± 4.92 | **39.90±3.86** |
| | 0.5 – 1 | 0.00 ± 0.00 | 0.00 ± 0.00 | 0.00 ± 0.00 | 0.00 ± 0.00 | 2.59 ± 1.14 | 3.84 ± 0.71 | 9.83 ± 0.85 | 9.04 ± 1.48 | **24.19±2.19** |
| 205 *more soluble in water* and *higher permeability* | 0 – 0 | 5.06 ± 1.21 | 3.53 ± 0.38 | 4.88 ± 2.21 | 8.17 ± 1.03 | 35.69 ± 3.19 | 39.74 ± 2.26 | **56.40 ± 4.15** | 18.87 ± 5.02 | 12.85±2.68 |
| | 0.5 – 10 | 1.16 ± 0.68 | 0.67 ± 0.55 | 0.66 ± 0.54 | 0.00 ± 0.00 | 19.15 ± 0.73 | 22.66 ± 1.90 | **39.22 ± 0.23** | 15.24 ± 1.63 | 10.44±5.75 |
| 206 *more soluble in water* and *lower permeability* | 0 – 0 | 12.17 ± 1.05 | 10.43 ± 2.88 | 13.08 ± 2.28 | 19.83 ± 2.46 | 44.35 ± 0.68 | 30.87 ± 0.62 | 54.87 ± 0.96 | 41.97 ± 0.87 | **65.33±2.16** |
| | 0.5 – 10 | 6.20 ± 0.64 | 6.23 ± 2.31 | 6.67 ± 0.53 | 4.83 ± 0.85 | 28.67 ± 2.22 | 20.06 ± 1.26 | 43.91 ± 1.77 | 35.20 ± 2.29 | **52.90±2.23** |

To sum up, for each conversation round, we request a drug $x_R$ similar to $x_c$, which will be updated at each conversation round. The $x_c$ and $x_R$ serve as two in-context pairs to feed into ChatDrug, *i.e.*, *"The output drug at round [c] is [$x_c$], which is wrong. We find a sequence [$x_R$] which is correct and similar. Can you help improve the edited results?"* An illustration of this conversation is in Figure 1.

## 4 EXPERIMENT

**Specifications for ChatDrug.** In this section, we verify the effectiveness of ChatDrug for drug editing on three types of drugs: small molecules, peptides, and proteins. Since ChatDrug is agnostic to LLMs, we select GALACTICA, Llama2 and ChatGPT as our backbone LLMs in our experiments, named ChatDrug-GALACTICA, ChatDrug-Llama2, and ChatDrug-Turbo, respectively. We introduce three types of drugs and five categories of tasks accordingly: tasks 1xx and 2xx are single- and multi-objective tasks for small molecules (each task further includes two subtasks w.r.t. two thresholds as will be discussed next), tasks 3xx and 4xx are single- and multi-objective editing tasks for peptides, and task 5xx is for single-objective protein editing. Due to the space limitation, please check Appendix F for the full list. Details of implementation and hyperparameters are in Appendix G.

### 4.1 TEXT-GUIDED MOLECULE PROPERTY EDITING

The first experiment is text-guided molecule editing or molecule optimization. We adopt 16 single-objective tasks and 12 multi-objective editing tasks from MoleculeSTM (Liu et al., 2022a). These tasks are about the high-level properties of small molecules, like solubility in water and permeability.

**Data**: Both the input molecules and retrieval DB are sampled from ZINC (Irwin et al., 2012): we sample 200 and 10K molecules (with SMILES strings) from ZINC as input molecules and retrieval DB, respectively. **Prompt**: The text prompt is *"Can you make molecule [SMILES placeholder] [task requirement]? The output molecule should be similar to the input molecule"*. The [task requirement] is the textual description for each specific task, *e.g.*, *more soluble in water* and *with higher permeability*. **Evaluation**. We take the hit ratio to measure the success ratio of edited molecules, *i.e.*, the percentage of edited molecules that can reach the desired properties compared to the input molecules. All the properties for small molecules considered here can be calculated deterministically using RDKit (Landrum et al., 2013). Another important argument is the threshold Δ: it is a successful hit if the difference between input and output properties is above the threshold.

Table 3: Visualization of six small molecule editing tasks. The blue regions , red regions , and green regions correspond to the edited substructures in the input molecule $x_{in}$, intermediate molecule $x_1$ for the 1st conversation round, and the output molecule $x_{out}$, respectively.

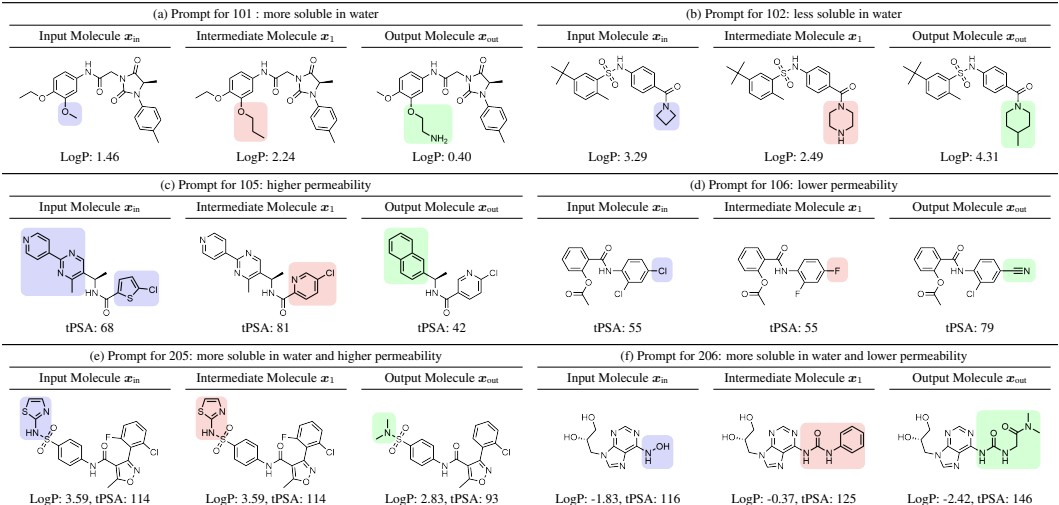

**Baselines**: The baselines (Liu et al., 2022a) are based on MegaMolBART (Irwin et al., 2022), a pretrained auto-regressive model. Baselines include Random, PCA, High-Variance, GS-Mutate, and MoleculeSTM with SMILES or Graph as the molecule representation.

**Observation.** We illustrate the descriptions and the single- and multi-objective editing results in Tables 1 and 2, respectively. The threshold $\Delta$ for each specific task is specified in Table 1; for multi-objective editing tasks in Table 2, the threshold $\Delta$ has two values corresponding to the two tasks. We further conduct an ablation study on the thresholds of ReDF in Section 4.5. We can observe that ChatDrug-Turbo can reach the best performance on 21 out of 28 tasks, 20 of which possess over 20% hit ratio than the second-best baseline method. We also note that ChatDrug-GALACTICA performs well, showing consistently higher hit ratios than baseline methods and reaching the best performance on 7 out of 28 tasks. However, ChatDrug-Llama2 is not as outstanding as the other ChatDrug models, outperforming on 7 out of 28 tasks. The reason may be that the corresponding backbone model Llama2 is not well pretrained on small molecule domain datasets. Table 3 visualizes 6 examples of molecule editing where ChatDrug-Turbo fails to generate molecules $x_1$ with desirable property change in the first conversation round, while successfully generates molecules $x_{out}$ ultimately. For example, in Table 3a, $x_1$ converts a methyl group to a propyl which incorrectly yields a less soluble molecule. Through conversational guidance, ChatDrug-Turbo changes its output $x_{out}$ to an aminoethyl group, successfully fulfilling the task. In Table 3f, $x_1$ installs a phenyl urea to the molecule, which brings lower permeability as requested but makes the molecule less soluble. In contrast, ChatDrug-Turbo is able to replace the hydrophobic aromatic substituent with a hydrophilic amide in $x_{out}$, consistent with the requirement of higher solubility in water.

## 4.2 TEXT-GUIDED IMMUNOGENIC BINDING PEPTIDE EDITING

The second task is text-guided immunogenic binding peptide editing. Immunogenic peptides are promising therapeutic targets for the personalized vaccine, which triggers a person's immune system, *e.g.*, CD8+ T cells, to fight diseases (Craiu et al., 1997; Hennecke & Wiley, 2001). Immunogenic peptides are typically degraded from intracellular antigens. To activate CD8+ T cell immune responses, these peptides must first bind to Major Histocompatibility Complex (MHC) proteins, forming peptide-MHC complexes which are then presented on the surface of infected or malignant cells to interact with the T cells. Although the peptide-MHC binding process is critical for immune response, it is highly specific, making editing known peptides to improve their binding affinity to specific MHC proteins a challenging yet crucial task for peptide engineering and discovery. Recall that peptides are typically short protein chains, with most peptides having less than 16 amino acids.

**Data**: In this experiment, we use the experimental dataset of peptide-MHC binding affinities (O'Donnell et al., 2020). This dataset contains 149 human MHC Class I proteins (alleles) and 309K peptides. We follow existing works (Chen et al., 2023) on using the 30 common MHC

Table 4: Results on six single-objective and three multi-objective peptide editing tasks. Random Mutation-$R$ for $R$ mutated positions. The evaluation is the hit ratio of the increased binding affinity score. The best results are marked in **bold**. Due to the space limitation, please check Appendix F for the text prompt of each task.

| | single-objective editing | | | | | | multi-objective editing | | |
|---|---|---|---|---|---|---|---|---|---|
| | 301 | 302 | 303 | 304 | 305 | 306 | 401 | 402 | 403 |
| Random Mutation-1 | 1.80 | 14.40 | 1.80 | 1.80 | 12.00 | 5.60 | 3.20 | 0.80 | 0.40 |
| Random Mutation-2 | 1.80 | 13.40 | 2.80 | 3.00 | 8.40 | 4.40 | 2.20 | 0.60 | 1.20 |
| Random Mutation-3 | 1.80 | 9.40 | 2.40 | 4.20 | 9.00 | 3.80 | 3.00 | 0.60 | 0.80 |
| ChatDrug-GALACTICA | 11.55 | 12.78 | 13.47 | 9.28 | 8.40 | 14.85 | 5.51 | 4.50 | 2.48 |
| ChatDrug-Llama2 | 27.64 | 14.89 | 21.18 | 13.79 | 19.52 | 26.33 | 8.33 | 4.95 | 3.80 |
| ChatDrug-Turbo | **56.60** | **69.80** | **64.33** | **59.04** | **65.00** | **64.13** | **44.69** | **34.54** | **41.77** |

(a) Motifs of input peptides for 301.    (b) Motifs of edited peptides for 301.    (c) Motifs of experimental peptides for 301.

(d) Motifs of input peptides for 302.    (e) Motifs of edited peptides for 302.    (f) Motifs of experimental peptides for 302.

Figure 2: Visualization of two peptide editing tasks using PWM. The x-axis corresponds to the position index, while the y-axis corresponds to the distribution of each amino acid (in alphabets) at each position.

proteins (alleles) and we randomly pick one as the source allele and one or more alleles as the target alleles. Notice that for single-allele tasks, 30 MHC proteins can be further divided into 3 categories: HLA-A, HLA-B, and HLA-C; we make sure that the sampled source and target alleles are from different categories. Then we sample 500 peptides from the source allele types. For the retrieval DB, the experimental data of the target allele(s) are adopted. The sampled MHC types are further specified in Appendix F. **Prompt**: We expect the edited peptides can bind to the target MHC protein(s), so the prompt template is *We want a peptide that binds to [target allele]. We have a peptide [peptide sequence] that binds to [source allele], can you help modify it? The output peptide should be similar to the input peptide."* **Evaluation**: The actual bindings require wet-lab experiments, which are expensive and prohibited for large scaled evaluation. Following existing works (Chen et al., 2021; 2023), we leverage the MHCflurry2.0 (O'Donnell et al., 2020) as a pseudo-oracle to predict the peptide-MHC binding affinity. MHCflurry2.0 is the state-of-the-art method enabling accurate estimating of the binding affinity of peptides with MHC proteins. The success of the peptide editing needs to satisfy two conditions: (1) The output peptide should have a higher binding affinity with the target allele compared to the input peptide; (2) The binding affinity of the output peptide and target allele should be above a certain threshold. Here we take the threshold as one-half of the average binding affinity of experimental data on the target allele. **Baselines**: Since there is no existing approach for text-guided binding peptide editing, we use random mutation as the baseline, *i.e.*, conducting random mutation on the amino acid sequence of the input peptides.

**Observation.** We illustrate the single- and multi-objective editing results in Table 4. We can observe that ChatDrug reaches the best performance over all 9 tasks compared to the random mutation baselines for all three LLMs backbones. ChatDrug-Turbo reaches the best performance with significantly higher hit ratios than ChatDrug-GALACTICA and ChatDrug-Llama2. We further visualize peptides generated by ChatDrug-Turbo using position weight matrices (PWMs) in Figure 2. PWM has been widely used for the visualization of protein motifs (patterns), and it plots the distribution of each amino acid at the corresponding position. Thus, more important motifs with higher probabilities will be marked in higher alphabets. According to Figure 2, the edited or optimized peptides follow similar patterns to the experimental data presented. For instance, for task 301, the edited peptides can successfully upweight the alphabet E (glutamic acid) at position 2. Results suggest a strong correlation between edited peptide binding motifs and those derived from experimental data.

## 4.3 TEXT-GUIDED PROTEIN SECONDARY STRUCTURE EDITING

Last but not least, we consider text-guided protein secondary structure editing (PSSE) (Klausen et al., 2019). For protein 1D sequence, it can fold into the 3D structure, as shown in Figure 1. Specifically, proteins possess four levels of structures, and secondary structures are fundamental building blocks, which are local folding patterns stabilized by hydrogen bonds. Typical secondary structures include

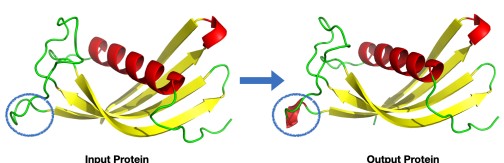 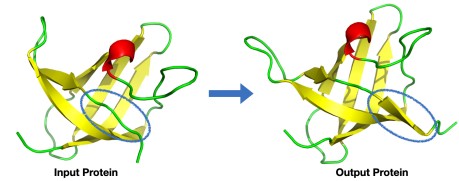

(a) Protein editing with more helix structures.  (b) Protein editing with more strand structures.

Figure 3: Visualization of two protein editing tasks. For the protein secondary structures, the $\alpha$-helix is marked in red, and $\beta$-sheet is marked in yellow. The edited regions before and after ChatDrug are marked in blue circles.

$\alpha$-helix and $\beta$-sheet, consisting of $\beta$-strands. Here we are interested in two PSSE tasks, *i.e.*, using ChatDrug to edit protein sequences with more helix or strand structures after folding.

**Data**: TAPE (Rao et al., 2019) is a benchmark for protein sequence property prediction, including the secondary structure prediction task. We take the test dataset and training dataset as the input proteins and retrieval DB, respectively. **Prompt**: *For an input protein sequence [protein sequence], can you modify it with more helix/strand structures?* **Baselines**: Same with peptide editing, we adopt random mutation as baselines. **Evaluation.** For evaluation, we adopt the state-of-the-art pretrained secondary structure prediction model, *i.e.*, ProteinCLAP-EBM-NCE from ProteinDT (Liu et al., 2023b). The hit condition is if the output protein sequences have more secondary structures than the input sequences.

**Observation.** As shown in Table 5, we can tell the large performance gain by ChatDrug, where ChatDrug-Llama2 and ChatDrug-Turbo achieve the highest hit ratio on task 501 and 502, respectively. We further visualize cases on how ChatDrug-Turbo successfully edits the proteins with more helix/strand structures. We adopt pretrained

Table 5: Results on two protein editing tasks. Random Mutation-$R$ for $R$ mutated positions. The evaluation is the hit ratio of increased secondary structures accordingly.

|  | 501 more helix | 502 more strand |
|---|---|---|
| Random Mutation-3 | 26.90 | 21.44 |
| ChatDrug-GALACTICA | 11.75 | 5.99 |
| ChatDrug-Llama2 | **34.79** | 34.79 |
| ChatDrug-Turbo | 33.18 | **59.68** |

ESMFold (Lin et al., 2022) for protein folding (protein sequence to protein structure prediction) and then plot the protein structures using PyMOL (Schrödinger & DeLano). We show two examples in Figure 3. As circled in the blue regions in Figures 3a and 3b, the edited proteins possess more helix structures and strand structures, respectively. More visualization can be found in Appendix H.

## 4.4 WHY CHATDRUG WORKS? KNOWLEDGE EXTRACTION

We are also interested in understanding how ChatDrug can work. As shown in Figure 4, we illustrate a case study on small molecule editing. It can be observed that ChatDrug can do knowledge extraction: for a specific task on editing molecules to be more soluble in water, ChatDrug can extract the reasonings and summarize them into five rules. This gives us the confidence that the success of ChatDrug is its ability of domain interpretation. We conduct further ablation studies like knowledge extraction without the context as a control experiment in Appendix I. Although ChatDrug can extract domain-specific information for the editing tasks, we

Figure 4: Knowledge extraction of ChatDrug.

notice a minor issue: the **redundancy** among knowledge. As shown in Figure 4, the extracted rules 1, 3, and 5 are all centered on introducing polar functional groups for solubility in water, despite from slightly different perspectives. In Appendix I, we continue to explore how ChatDrug can play a positive role in knowledge summarization, aiming to alleviate this knowledge redundancy issue.

## 4.5 ABLATION STUDIES

**Ablation Study on Comparison with Zero-shot and In-context Learning.** There are two important modules for ChatDrug: conversation for result refinement and the ReDF for knowledge retrieval. Thus in this ablation study, we would like to explore the effect of such two modules. The first case is zero-shot prediction. It is indeed ChatDrug with $c = 0$, *i.e.*, without conversation or ReDF. On the other hand, in-context learning (ICL) can be treated as ChatDrug equipped with the ReDF module but without any conversational round. Concretely, the retrieved drug is $\boldsymbol{x}_R = \text{ReDF}(\boldsymbol{x}_{\text{in}}, \boldsymbol{x}_{\text{in}})$, with $\tilde{x} =$

Table 6: Ablation studies on comparison with in-context learning (ICL) and conversation rounds on molecule editing. The threshold is the loose threshold with $\Delta = 0$, and the random seed is 0.

| | $C$ | 101 | 102 | 103 | 104 | 105 | 106 | 107 | 108 | 201 | 202 | 203 | 204 | 205 | 206 |
|---|---|---|---|---|---|---|---|---|---|---|---|---|---|---|---|
| ICL (few-shot) | | 52.11 | 75.45 | 37.76 | 46.23 | 30.64 | 42.86 | 54.97 | 69.81 | 59.88 | 39.86 | 53.45 | 49.36 | 37.42 | 42.77 |
| ChatDrug-Turbo | $C = 0$ (zero-shot) | 78.26 | 71.35 | 16.15 | 32.12 | 16.04 | 8.33 | 59.41 | 63.16 | 43.09 | 0.52 | 54.49 | 0.53 | 2.11 | 22.22 |
| | $C = 1$ | 89.56 | 93.64 | 48.35 | 61.62 | 47.93 | 56.97 | 90.00 | 93.08 | 72.29 | 36.26 | 86.14 | 30.00 | 9.44 | 54.14 |
| | $C = 2$ | 93.37 | 97.11 | 52.81 | 67.93 | 55.76 | 78.40 | 95.57 | 98.10 | 80.37 | 48.52 | 90.18 | 39.88 | 12.72 | 67.23 |
| | $C = 3$ | 96.11 | 97.69 | 55.11 | 75.54 | 59.51 | 87.65 | 98.09 | 98.73 | 83.75 | 60.49 | 92.02 | 50.32 | 15.48 | 76.74 |
| | $C = 4$ | 96.67 | 97.69 | 59.20 | 78.14 | 63.35 | 94.41 | 98.09 | 98.73 | 86.79 | 68.32 | 94.41 | 57.42 | 22.36 | 80.00 |
| | $C = 5$ | 97.22 | 97.69 | 59.77 | 83.06 | 65.84 | 95.03 | 99.36 | 98.73 | 89.17 | 70.19 | 94.41 | 63.40 | 25.32 | 81.55 |

Table 7: Ablation studies on thresholds in domain feedback function $D$ with two conversational rounds. The evaluation function $E$ uses the strict threshold. We report the mean of five seeds, and stds are in Appendix I.

| | 101 | 102 | 103 | 104 | 105 | 106 | 107 | 108 | 201 | 202 | 203 | 204 | 205 | 206 |
|---|---|---|---|---|---|---|---|---|---|---|---|---|---|---|
| loose threshold | 80.73 | 41.00 | 11.23 | 16.94 | 33.16 | 53.59 | 14.96 | 21.93 | 20.14 | 7.96 | 17.93 | 5.79 | 3.66 | 41.04 |
| strict threshold | 88.67 | 70.08 | 19.37 | 30.99 | 43.08 | 66.69 | 72.60 | 76.43 | 49.64 | 24.92 | 53.64 | 24.19 | 10.44 | 52.9 |

(a) Task 101 *more soluble in water*  (b) Task 102 *less soluble in water*  (c) Task 107 *more hydrogen bond acceptors*

Figure 5: Similarity distribution between input molecules $x_{\text{in}}$ and retrieval $x_R$, intermediate $x_1$, and output molecules $x_{\text{out}}$. We pick up three tasks on small molecules for visualization, and more results are in Appendix H.

$x_{\text{in}}$ in Equation (2). The text prompt for zero-shot and ICL are *"Can you edit the molecule [$x_{in}$] to be more soluble?"* and *''We know that [$x_R$] is similar to [$x_{in}$] and is more soluble in water. Can you edit the molecule [$x_{in}$] to be more soluble?"* The results are in Table 6, and we observe that both ChatDrug and ICL are better than the zero-shot, and conversational refinement performs best on all 14 tasks.

**Ablation Study on the Number of Conversation Rounds in ChatDrug.** In ChatDrug, the number of conversation rounds is an important hyperparameter. Here we conduct an ablation study on small molecules to test its effectiveness. The results are in Table 6. For molecule editing tasks tested here, the performance of ChatDrug-Turbo tends to converge after $C = 2$ conversation rounds.

**Ablation Study on the Thresholds in Feedback Condition Function.** In ChatDrug, another important factor is the domain feedback function $D(\cdot, \cdot; \cdot)$. For molecule editing, we discuss two thresholds when evaluating with $E(\cdot, \cdot; \cdot)$. One is $\Delta = 0$ (loose condition), and the other is $\Delta > 0$ (strict condition), where the $\Delta$ value is different for each task. Here we conduct ablation studies on two conditions for feedback function $D$. The results are in Table 7, and the observation is that ChatDrug-Turbo with the stricter threshold in feedback condition can lead to higher accuracy by a large margin. Note that for each task in Tables 1 and 2, we keep the same threshold for $D$ and $E$.

**Ablation Study on the Similarity Between Input and Output Drugs.** We plot the distribution of similarities between input molecules $x_{\text{in}}$ and retrieval $x_R$, intermediate $x_1$, and output molecules $x_{\text{out}}$ using ChatDrug-Turbo. The similarity distributions of three tasks are in Figure 5, and more results are in Appendix H. One interesting observation is that the similarities between $x_{\text{in}}$ and intermediate molecules are quite $x_1$, but the hit ratio is the lowest among the three. Then we plot the similarity $x_{\text{in}}$ and $x_R$, where the similarities are comparatively low, yet the hit ratio is the highest. This reveals that the ReDF module can explore the chemical space to search for more optimal solutions. Then, by utilizing such retrieved information and repeating $C$ conversational rounds, ChatDrug will do a trade-off between the similarity with input molecules $x_{\text{in}}$ and knowledge explorations.

## 5 LIMITATION AND CONCLUSION

In this work, we present ChatDrug, a framework that utilizes LLMs for drug editing tasks. We build up a benchmark on 39 tasks over three main types of drugs: small molecules, peptides, and proteins. Empirical results have verified the effectiveness of ChatDrug on these drug editing tasks, and the visual analysis further qualitatively illustrates how ChatDrug can modify the key substructures for the target properties. Meanwhile, ChatDrug also possesses limitations. One limitation is that ChatDrug is bad at understanding the complex structures of drugs, *i.e.*, the 3D geometries. This may require a more profound utilization of geometric modeling. Another limitation is that ChatDrug requires certain conversational rounds to reach strong performance. An ideal solution is to reduce such computational costs using the knowledge summarization ability of LLMs, and we leave this for future exploration.

## REPRODUCIBILITY STATEMENT

The codes and datasets can be found at this GitHub link. Additionally, we provide a detailed description of ChatDrug in Section 3. All the datasets used in our experiments can be found in each subsection of Section 4. All prompts we used in ChatDrug are attached in Appendix F. We also present our implementation settings and hyperparameters in Appendix G.

## ACKNOWLEDGEMENT

Jiongxiao Wang and Chaowei Xiao are supported by the Department of Homeland Security under Grant Award 17STQAC00001-06-00.

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

## A    THE UTILIZATION OF CHATGPT AS BACKBONE LLM

Some concerns have been raised on the utilization of using ChatGPT as the backbone LLM. We want to explain this from multiple perspectives.

(1) Our ChatDrug is agnostic to the backbone LLMs, and we test in-total three backbone LLMs: Turbo (ChatGPT), GALACTICA, and Llama2. Among these, ChatDrug-Turbo achieves the best performance and highest stability for all three drug types. For ChatDrug on the open-source LLMs, ChatDrug-GALACTICA only performs better than baseline on the small molecule editing and peptide editing, but cannot deal with protein editing tasks. ChatDrug-Llama2 can reach better performance than baseline methods on peptide editing and protein editing but fails to cover small molecule editing tasks. Such unstable performance across different drug types may be attributed to the limited domain-specific training data for open-source LLMs. However, the discussions about training LLMs are beyond the scope of this paper.

(2) In general, ChatGPT has been widely used in both the machine learning and domain (chemistry and biology) community. For example:

- In domain journals: scientific papers that use ChatGPT for discovery have been published in top journals such as PNAS (ChatGPT Outperforms Crowd-Workers for Text-Annotation Tasks (Gilardi et al., 2023)), Radiology (A Context-based Chatbot Surpasses Trained Radiologists and Generic ChatGPT in Following The ACR Appropriateness Guidelines (Rau et al., 2023)), Nature Medicine (Large Language Models in Medicine (Thirunavukarasu et al., 2023)), and Nature Chemistry (ChatGPT Saves The Day (Francl, 2023)).
- In machine learning communities: many machine learning papers with ChatGPT usage have been recently accepted by NeurIPS 2023 including HuggingGPT: Solving AI Tasks with ChatGPT and its Friends in Hugging Face (Shen et al., 2023), Mathematical Capabilities of ChatGPT (Frieder et al., 2023) and ChatGPT-Powered Hierarchical Comparisons for Image Classification.

## B    EXAMPLES OF CHATDRUG

Here we show three concrete examples of the editing tasks on small molecules, peptides, and proteins.

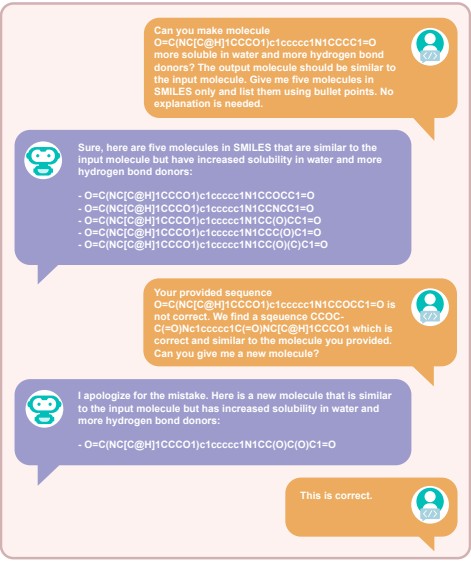

Figure 6: Example of small molecule editing (task 203).

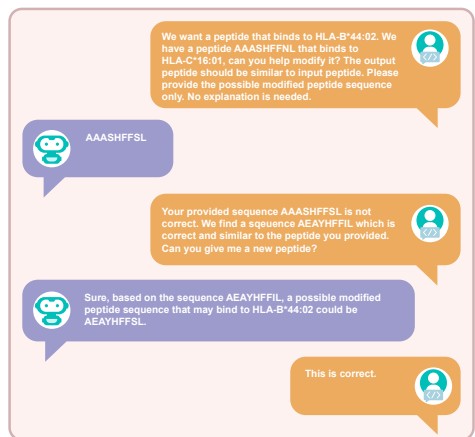

Figure 7: Example of peptide editing (task 301).

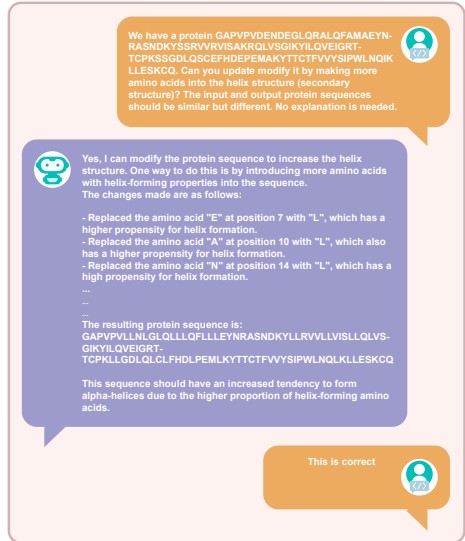

Figure 8: Example of protein editing (task 501).

## C    MORE DISCUSSIONS ON CHATDRUG

In this section, we will discuss three aspects of ChatDrug and the general conversational LLMs for drug discovery: the scope, the main attributes, and the key challenges and guidelines when using ChatDrug.

**Scope of ChatDrug.** The first natural question raised is *What are the feasible drug discovery tasks for ChatDrug?* To answer this question, we need to reiterate the conversational LLM's feasibility for drug discovery tasks. There have been a series of works (Zeng et al., 2022; Edwards et al., 2022; 2021; Taylor et al., 2022; Su et al., 2022) exploring the LLMs for small molecule and protein discovery, ranging from molecule representation to text-to-molecule generation. These are important machine learning tasks, revealing domain data's (conditional) distribution learning ability. However, there exists another important task in real scenarios: *drug editing* (a.k.a. *lead optimization* or *protein design* in domain applications). This is a routine task in pharmaceutical companies, and it aims at updating the molecule's substructures (Mihalić & Trinajstić, 1992), related to certain key tactics in drug discovery like functional group change (Ertl et al., 2020) and scaffold hopping (Böhm et al., 2004; Hu et al., 2017). Traditional solutions relying on domain experts for manual editing can be subjective or biased (Drews, 2000; Gomez, 2018). To alleviate this issue, multi-modal models with LLMs provide a promising solution, and recent works (Liu et al., 2023b; 2022a) have started to explore text-guided drug editing or controllable drug generation. However, these works are domain-specific (*e.g.*, only for small molecules or proteins) and do not possess conversational potentials like ChatGPT. In contrast, ChatDrug possesses outstanding generalization abilities to various domain tasks and enables conversational refinement in drug editing tasks. [2]

**Attributes of ChatDrug.** We conclude three fundamental attributes of ChatDrug: open vocabulary, compositionality, and inspiration. (1) Open vocabulary means ChatDrug is not limited to a fixed set of pre-defined drug-related annotations. Instead, it can generalize to novel drug concepts due to the unbound vocabulary depicted by the natural language. (2) Compositionality means we can express a complex concept by decomposing it into simple concepts. (3) Inspiration means the position of ChatDrug is to inspire domain experts with knowledge exploration but not replace them. A concrete example illustrating these three attributes is *multi-objective lead optimization*. We can use natural language to guide us to generate an entirely new attribute of a molecule (open vocabulary); meanwhile, the new attribute is composed of multiple simple attributes, such as binding to a new protein and high permeability (compositionality). Finally, such an optimized molecule may not be directly used for real scenarios, but it can provide insights for domain experts in drug design (inspiration).

**Challenges and Guidelines when Using ChatDrug.** Now that we have decided to narrow ChatDrug to the drug editing tasks, we need to scrutinize more details before deploying it. With careful reconsidering, we summarize two main challenges that we need to keep in mind. (1) ChatDrug can do better in fuzzy searching than exact searching in drug editing tasks. Drug editing tasks, or drug controllable generation, can cover various topics. However, one critical difference between ChatDrug and other LLMs in vision tasks is that ChatDrug or drug discovery is **a scientific problem** while the image and video (Radford et al., 2021; Nichol et al., 2021; Ramesh et al., 2022; Patashnik et al., 2021; Fan et al., 2022) generation is more of **an artistic endeavor**. Namely, for text prompts like "I want to add an isobutyl group on the 3-position of the aromatic ring in Aspirin", domain experts can do this precisely, and thus the impact of ChatDrug is limited here. However, for other tasks like "I want to modify this molecule to be more soluble in the water", the results are not deterministic, and this is where LLMs can act as a more useful tool to inspire the domain experts. These two types of text prompts are called exact searching and fuzzy searching, respectively. We conclude that ChatDrug is more beneficial for the fuzzy searching problem. (2) ChatDrug relies on the pretrained LLMs, initially pretrained on a large-scale and universal corpus. Thus, there is a noticeable domain shift when applying them to domain-specific tasks. However, as will be shown in Sections 3 and 4, the existing LLMs illustrate the interpretation ability of the domain knowledge. Though such interpretation is preliminary, we believe that ChatDrug is an inspiring and promising direction for future usage in both communities.

---

[2]We acknowledge that there have been certain parallel works (Bran et al., 2023; Boiko et al., 2023) exploring conversational LLMs on reaction and synthesis tasks.

# D    RELATED WORK

## D.1    MULTI-MODAL MODELING FOR SMALL MOLECULE DISCOVERY

Small molecules can be roughly categorized into two big modalities (Zeng et al., 2022; Liu et al., 2022a): the **internal chemical structure** and **external description**. The internal chemical structure refers to the molecule's structure information, *e.g.*, 1D sequence (SMILES), 2D molecular graph, and 3D geometric graph. On the other hand, the external description depicts the high-level information of molecules, *e.g.*, the molecule's binding affinity with potential targets, and the functionalities of molecules.

Recently, a research line has been starting to bridge the gap between such two modalities. KV-PLM (Zeng et al., 2022) first applies the joint masking auto-encoding on the SMILES string and biomedical textual description. Text2Mol (Edwards et al., 2021) conducts contrastive learning between molecular graph and text data for retrieval tasks between modalities. MolT5 (Edwards et al., 2022) does the translation between SMILES and textual annotation of molecules in a mutual way. MoMu (Su et al., 2022) also conducts contrastive learning, while it considers both the retrieval and molecule captioning and text-to-molecule tasks. MoleculeSTM (Liu et al., 2022a) proposes a larger molecule-text dataset and highlights the text-guided molecule editing tasks. Such tasks reveal the potential of LLMs for more realistic drug discovery tasks.

## D.2    MULTI-MODAL MODELING FOR PEPTIDE AND PROTEIN DISCOVERY

There have also been several works exploring multi-modal modeling for protein discovery. Pro-Gen (Madani et al., 2020) is a text-to-sequence protein design framework, but it is fixed to a predefined set of texts, which can be treated with indices. Thus it is not open-vocabulary and lacks the generalization ability to novel textual descriptions. Besides, the predefined texts and indices cannot sufficiently describe the protein functions (Zhang et al., 2020). ProteinDT (Liu et al., 2023b) is a recent work that addresses this issue with the free-text protein design. A parallel work is Chroma (Ingraham et al., 2022), and it conducts text-guided protein editing on the backbone structure instead of the sequence.

# E   DATA SPECIFICATION

Drugs like small molecules and proteins can have multiple modalities. Specifically, small molecules can be naturally represented as 1D sequence, 2D molecular graph, and 3D geometric graph, biological knowledge graph, and textual description. The first three data structures capture the internal chemical structure information, while the last two data structures provide a higher-level view of the molecule's functionalities (*e.g.*, the molecule's interactions with other proteins or diseases.).

There are 20 amino acids in nature, as listed below:

Table 8: 20 amino acids and the corresponding abbreviations.

| Amino Acid | Alphabet |
| --- | --- |
| Isoleucine | I |
| Valine | V |
| Leucine | L |
| Phenylalanine | F |
| Cysteine | C |
| Methionine | M |
| Alanine | A |
| Glycine | G |
| Threonine | T |
| Serine | S |
| Tryptophan | W |
| Tyrosine | Y |
| Proline | P |
| Histidine | H |
| Asparagine | N |
| Asparatic acid | D |
| Glutamine | Q |
| Glutamic acid | E |
| Lysine | K |
| Arginine | R |

## F    TASK SPECIFICATION

Here we present all the task specifications and prompts used in our experiments.

- For ChatDrug-Turbo and ChatDrug-Llama2, we list the template of prompts of two stages of PDDS and ReDF in Tables 9, 12 and 15 for small molecules, peptides, and proteins, respectively.
- Different from ChatDrug-Turbo and ChatDrug-Llama2, the base model of ChatDrug-GALACTICA, GALACTICA-6.7b, is not instruction tuned. Besides, GALACTICA can support Question Answering tasks and has additional special tokens for marking the start and the end of molecules and protein annotations. Such differences make us to choose a different template of prompts specifically designed for ChatDrug-GALACTICA, which are shown in Tables 10, 13 and 16
- We list the corresponding task requirement and allele type information in Tables 11, 14 and 17.
- We further list the prompts of in-context learning in Table 18 for reference.

Table 9: Prompt for small molecule editing. The task requirement can be found in Table 11.

| Task | Module | Prompt |
|---|---|---|
| 1xx (101-108) | PDDS | Can you make molecule [input SMILES] [task requirement 1]? The output molecule should be similar to the input molecule. Give me five molecules in SMILES only and list them using bullet points. No explanation is needed. |
| | ReDF | Your provided sequence [output SMILES] is not correct. We find a sequence [retrieved SMILES] which is correct and similar to the molecule you provided. Can you give me a new molecule? |
| 2xx (201-206) | PDDS | Can you make molecule [input SMILES] [task requirement 1] and [task requirement 2]? The output molecule should be similar to the input molecule. Give me five molecules in SMILES only and list them using bullet points. No explanation is needed. |
| | ReDF | Your provided sequence [output SMILES] is not correct. We find a sequence [retrieved SMILES] which is correct and similar to the molecule you provided. Can you give me a new molecule? |

Table 10: Prompt for small molecule editing with ChatDrug-GALACTICA. The task requirement can be found in Table 11.

| Task | Module | Prompt |
|---|---|---|
| 1xx (101-108) | PDDS | Question: Can you make molecule [START_I_SMILES][input SMILES][END_I_SMILES] [task requirement 1]? The output molecule should be similar to the input molecule.\n |
| | ReDF | Question: Your provided sequence [START_I_SMILES][output SMILES][END_I_SMILES] is not correct. We find a sequence [START_I_SMILES][retrieved SMILES][END_I_SMILES] which is correct and similar to the molecule you provided. Can you give me a new molecule?\n |
| 2xx (201-206) | PDDS | Question: Can you make molecule [START_I_SMILES][input SMILES][END_I_SMILES] [task requirement 1] and [task requirement 2]? The output molecule should be similar to the input molecule.\n |
| | ReDF | Question: Your provided sequence [START_I_SMILES][output SMILES][END_I_SMILES] is not correct. We find a sequence [START_I_SMILES][retrieved SMILES][END_I_SMILES] which is correct and similar to the molecule you provided. Can you give me a new molecule?\n |

Table 11: Task requirement for small molecule editing, corresponding to Table 9 and Table 10.

| Task ID | Task Requirement 1 | Task Requirement 2 |
|---|---|---|
| 101 | more soluble in water | None |
| 103 | more like a drug | None |
| 104 | less like a drug | None |
| 105 | higher permeability | None |
| 106 | lower permeability | None |
| 107 | more hydrogen bond acceptors | None |
| 108 | more hydrogen bond donors | None |
| 201 | more soluble in water | more hydrogen bond acceptors |
| 202 | less soluble in water | more hydrogen bond acceptors |
| 203 | more soluble in water | more hydrogen bond donors |
| 204 | less soluble in water | more hydrogen bond donors |
| 205 | more soluble in water | higher permeability |
| 206 | more soluble in water | lower permeability |

Table 12: Prompt for peptide editing. The source allele target type and target allele type can be found in Table 14.

| Task | Stage | Prompt |
|---|---|---|
| 3xx (301-306) | PDDS | We want a peptide that binds to [target allele type 1]. We have a peptide [input peptide] that binds to [source allele type], can you help modify it? The output peptide should be similar to input peptide. Please provide the possible modified peptide sequence only. No explanation is needed. |
| | ReDF | Your provided sequence [output peptide] is not correct. We find a sequence [retrieved peptide] which is correct and similar to the peptide you provided. Can you give me a new peptide? |
| 4xx (401-403) | PDDS | We want a peptide that binds to [target allele type 1] and [target allele type 2]. We have a peptide [input peptide] that binds to [source allele type], can you help modify it? The output peptide should be similar to input peptide. Please provide the possible modified peptide sequence only. No explanation is needed. |
| | ReDF | Your provided sequence [output peptide] is not correct. We find a sequence [retrieved peptide] which is correct and similar to the peptide you provided. Can you give me a new peptide? |

Table 13: Prompt for peptide editing with ChatDrug-GALACTICA. The source allele target type and target allele type can be found in Table 14.

| Task | Stage | Prompt |
|---|---|---|
| 3xx (301-306) | PDDS | Question: We want a peptide that binds to [target allele type 1]. We have a peptide [input peptide] that binds to [source allele type], can you help modify it? The output peptide should be similar to input peptide.\n |
| | ReDF | Question: Your provided sequence [output peptide] is not correct. We find a sequence [retrieved peptide] which is correct and similar to the peptide you provided. Can you give me a new peptide?\n |
| 4xx (401-403) | PDDS | Question: We want a peptide that binds to [target allele type 1] and [target allele type 2]. We have a peptide [input peptide] that binds to [source allele type], can you help modify it?\n |
| | ReDF | Question: Your provided sequence [output peptide] is not correct. We find a sequence [retrieved peptide] which is correct and similar to the peptide you provided. Can you give me a new peptide?\n |

Table 14: Target allele type and source allele type for peptide editing, corresponding to Table 12 and Table 13

| Task ID | Source Allele Type | Target Allele Type 1 | Target Allele Type 2 |
|---|---|---|---|
| 301 | HLA-C*16:01 | HLA-B*44:02 | None |
| 302 | HLA-B*08:01 | HLA-C*03:03 | None |
| 303 | HLA-C*12:02 | HLA-B*40:01 | None |
| 304 | HLA-A*11:01 | HLA-B*08:01 | None |
| 305 | HLA-A*24:02 | HLA-B*08:01 | None |
| 306 | HLA-C*12:02 | HLA-B*40:02 | None |
| 401 | HLA-A*29:02 | HLA-B*08:01 | HLA-C*15:02 |
| 402 | HLA-A*03:01 | HLA-B*40:02 | HLA-C*14:02 |
| 403 | HLA-C*14:02 | HLA-B*08:01 | HLA-A*11:01 |

Table 15: Prompt of Conversation Module for protein editing. The task requirement can be found in Table 17.

| Task ID | | Prompt |
|---|---|---|
| 5xx (501-502) | PDDS | We have a protein [input protein]. Can you update modify it by [task requirement]? The input and output protein sequences should be similar but different. No explanation is needed. |
| | ReDF | Your provided sequence [output protein] is not correct. We find a sequence [retrieved protein] which is correct and similar to the protein you provided. Can you give me a new protein? |

Table 16: Prompt of Conversation Module for protein editing with ChatDrug-GALACTICA. The task requirement can be found in Table 17.

| Task ID | | Prompt |
|---|---|---|
| 5xx (501-502) | PDDS | Question: We have a protein [START_AMINO][input protein][END_AMINO]. Can you update modify it by [task requirement]? The input and output protein sequences should be similar but different.\n |
| | ReDF | Question: Your provided sequence [START_AMINO][output protein][END _AMINO] is not correct. We find a sequence [START_AMINO][retrieved protein][END_AMINO] which is correct and similar to the protein you provided. Can you give me a new protein?\n |

Table 17: Task requirement for protein editing, corresponding to Table 15 and Table 16.

| Task ID | Task Requirement |
|---|---|
| 501 | making more amino acids into the helix structure (secondary structure) |
| 502 | making more amino acids into the strand structure (secondary structure) |

Table 18: Prompt of in-context learning.

| Task | Prompt |
|---|---|
| 1xx (101-108) | Can you make molecule [input SMILES] [task requirement]? The output molecule should be similar to the input molecule. We have known that similar molecule [retrieved SMILES] is one of the correct answers. Give me another five molecules in SMILES only and list them using bullet points. No explanation is needed. |
| 2xx (201-208) | Can you make molecule [input SMILES] [task requirement 1] and [ask requirement 2]? The output molecule should be similar to the input molecule. We have known that similar molecule [retrieved SMILES] is one of the correct answers. Give me another five molecules in SMILES only and list them using bullet points. No explanation is needed. |
| 3xx (301-306) | We want a peptide that binds to [target allele type]. We have a peptide [input peptide] that binds to [source allele type], can you help modify it? The output peptide should be similar to input peptide. We have known that similar peptide [retrieved peptide] is one of the correct answers. Please provide another possible modified peptide sequence only. No explanation is needed. |
| 4xx (401-403) | We want a peptide that binds to [target allele type 1] and [target allele type 2]. We have a peptide [input peptide] that binds to [source allele type], can you help modify it? The output peptide should be similar to input peptide. We have known that similar peptide [retrieved peptide] is one of the correct answers. Please provide another possible modified peptide sequence only. No explanation is needed. |
| 5xx (501-502) | We have a protein [input protein]. Can you update modify it by [task requirement]? The input and output protein sequences should be similar but different. We have known that similar protein [retrieved protein] is one of the correct answers. Please provide another possible modified protein only. No explanation is needed. |

## G  IMPLEMENTATION AND HYPERPARAMETERS

### G.1  CHATGPT-TURBO SETTINGS

We implement our experiments with ChatGPT-Turbo through OpenAI API. Specifically, we utilize the model $gpt$-3.5-$turbo$-0301 under $ChatCompletion$ function, which is a snapshot of $gpt$-3.5-$turbo$ from March 1st 2023. This model will not receive updates, so we can ensure the reproducibility of our results. We also set the $temperature$ to 0 to reduce the potential randomness in our experiments. Additionally, we observe that ChatGPT often generates repeated sequences or fails to stop generating sequences for chemistry-related questions. To mitigate this issue, we set the $frequency\_penalty$ to 0.2. Moreover, for improved adaptation to different domains, it is advisable to incorporate a system role prompt within ChatGPT. In our case, we utilize the following prompt: "You are an expert in the field of molecular chemistry."

### G.2  OPEN SOURCE LLMS USED WITHIN CHATDRUG

For ChatDrug-GALACTICA and ChatDrug-Llama2, we used the open source model checkpoint through Hugging Face. Corresponding model names in Hugging Face are $facebook/galactica$-6.7$b$ for ChatDrug-GALACTICA and $meta$-$llama/Llama$-2-7$b$-$chat$-$hf$ for ChatDrug-Llama2. For implementation, we set all parameters as default parameters.

### G.3  EXPERIMENTS THRESHOLD FOR SMALL MOLECULE EDITING

Following MoleculeSTM (Liu et al., 2022a), in our small molecule editing experiments, we utilize two different threshold settings: a loose threshold and a strict threshold. For the main results in Tables 1 and 2, we keep the same threshold for domain feedback function $D$ and evaluation function $E$. The threshold $\Delta$ used for each small molecule editing task is shown in Table 19, which holds for both functions.

Table 19: Threshold $\Delta$ for each small molecule editing task, $\Delta_1$ and $\Delta_2$ represent the threshold of task requirement 1 and task requirement 2, respectively.

| Task ID | Loose Threshold | | Strict Threshold | |
|---|---|---|---|---|
| | $\Delta_1$ | $\Delta_2$ | $\Delta_1$ | $\Delta_2$ |
| 101 | 0 | – | 0.5 | – |
| 102 | 0 | – | 0.5 | – |
| 103 | 0 | – | 0.1 | – |
| 104 | 0 | – | 0.1 | – |
| 105 | 0 | – | 10 | – |
| 106 | 0 | – | 10 | – |
| 107 | 0 | – | 1 | – |
| 108 | 0 | – | 1 | – |
| 201 | 0 | 0 | 0.5 | 1 |
| 202 | 0 | 0 | 0.5 | 1 |
| 203 | 0 | 0 | 0.5 | 1 |
| 204 | 0 | 0 | 0.5 | 1 |
| 205 | 0 | 0 | 0.5 | 10 |
| 206 | 0 | 0 | 0.5 | 10 |

### G.4  EXPERIMENTS THRESHOLD FOR PEPTIDE EDITING

For the peptide editing task, as mentioned in Section 4, we take the threshold as one-half of the average binding affinity of experimental data on the target allele. The original average binding affinity of each experimental data can be found in the source code.

### G.5 EVALUATION METRIC

We evaluate the performance of ChatDrug by hit ratio, which is computed by the following equation:

$$\text{Hit Ratio} = \frac{\text{Number of Success Sequence Editing}}{\text{Number of Valid Sequence Editing}} \tag{3}$$

One point we need to highlight is that if ChatDrug returns an invalid sequence, we would just skip and do not consider it in computing the hit ratio. That is why we use "Number of Valid Sequence Editing" as the denominator here.

In small molecule editing tasks, ChatDrug tends to return more than one sequence in the PDDS module. Thus, we add a prompt "Give me five molecules in SMILES only and list them using bullet points." to unify the numbers and format of molecules returned by ChatDrug. In the experiments of the Conversation module, we always choose the first valid molecule as the beginning of the conversation. We further carry out an ablation study to explore the effect of using more molecules in the PDDS module.

### G.6 RANDOMNESS

The experiment results of the PDDS Module are entirely deterministic. Any randomness observed in ReDF Module and Conversation Module is due to the utilization of different seeds during the sampling of retrieval database DB from ZINC for molecule editing.

Specifically, for small molecule editing, we adopt seed 0,1,2,3,4 for main results in Tables 1 and 2, and seed 0 for the other ablation studies.

### G.7 COMPUTATIONAL RESOURCES

All of our experiments for ChatDrug-Turbo are conducted on a single NVIDIA RTX A5000 GPU. The GPU is only used for peptide and protein evaluation. The primary cost incurred during our experiments comes from the usage of the OpenAI API for ChatGPT, which amounted to less than $100 in total.

For open source LLMs backbones, ChatDrug-GALACTICA and ChatDrug-Llama2 need at least 2 NVIDIA RTX A5000 GPUs for small molecule editing and peptide editing. For protein editing tasks, due to an extra usage of GPU for protein evaluation, 4 NVIDIA RTX A5000 GPUs are needed in our experiments.

# H QUALITATIVE ANALYSIS

In the main body, we provide 10 case studies and 3 similarity distributions to illustrate the effectiveness of ChatDrug for small molecule editing, peptide editing, and protein editing.

In this section, we provide additional case studies and similarity distributions as follows:

- We list 8 case studies on functional group change of small molecules in Appendix H.1.1.
- We list 14 similarity comparisons on small molecules in Appendix H.1.2.
- We list 9 motif updates for all 9 peptide editing tasks in Appendix H.2.
- We list 8 case studies on secondary structure change of proteins in Appendix H.3.

We want to specify that for all the qualitative analyses listed here, we are using $C = 2$ conversation rounds. Especially for small molecules, we consider random seed with 0 and the loose threshold, *i.e.*, $\Delta = 0$ for all tasks.

## H.1 SMALL MOLECULES

### H.1.1 FUNCTIONAL GROUP CHANGE ON SMALL MOLECULES

Table 20 visualizes examples of 8 molecule editing tasks where ChatDrug-Turbo successfully generates output molecules $x_{\text{out}}$ with desirable property change, while the output of the first conversation round $x_1$ fail. In Table 20a and b, $x_{\text{out}}$ successfully adds the desirable fragments to alter the drug likeness of $x_{\text{in}}$, while $x_1$ does so in the wrong direction. In Table 20c, $x_1$ installs a chloride but maintains the same number of hydrogen bond acceptors (HBAs). In contrast, ChatDrug-Turbo adds a salicylamide moiety that brings two more HBAs. Similarly, in Table 20d, the number of hydrogen bond donors (HBDs) remains in $x_1$ but successfully increases in $x_{\text{out}}$ via insertions of alcohols and amines.

In Table 20e and f, both cases of $x_1$ are able to increase the number of HBAs as indicated in the prompt, but the water solubilities shift oppositely. The output molecules successfully fix the trend. In particular, hydrophibicity is appropriately employed in Table 20f to balance the additional polarity from HBAs, generating a less soluble molecule. In Table 20g and h, both cases of $x_1$ satisfy the solubility requirement but not through the change of HBDs. In $x_{\text{out}}$, the problems are solved by having extra HBDs with further enhanced solubility changes.

Table 20: Visualization of additional eight small molecule editing cases. The blue regions , red regions , and green regions correspond to the edited substructures in the input molecule $x_{in}$, intermediate molecule $x_1$ in the 1st conversation round, and the output molecule $x_{out}$, respectively.

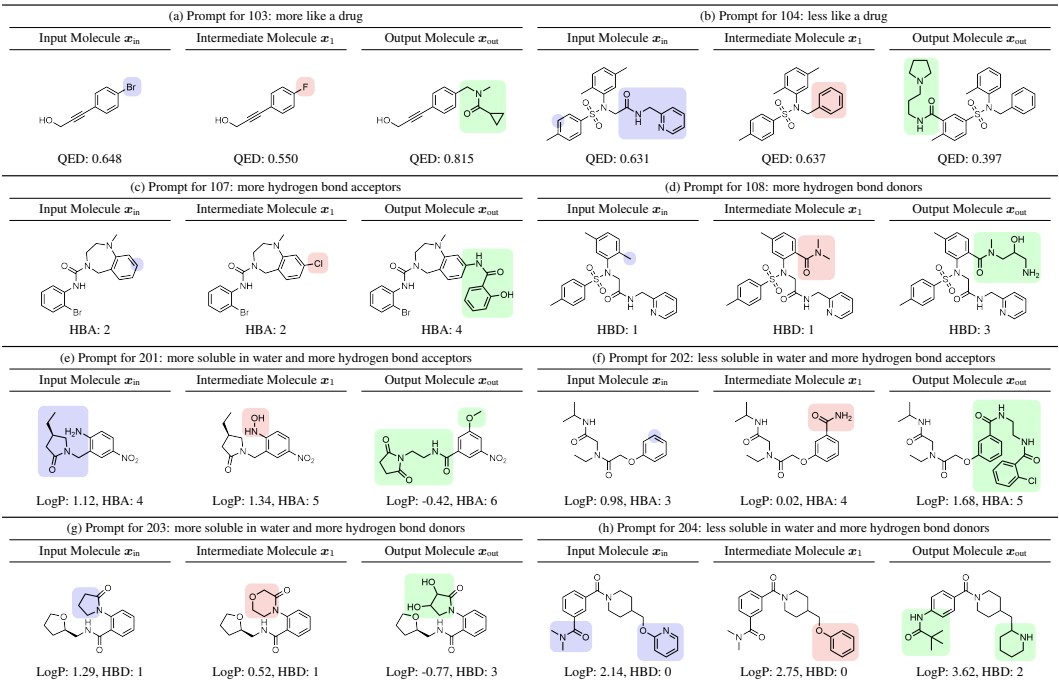

### H.1.2 SIMILARITY BETWEEN INPUT, INTERMEDIATE, RETRIEVED, AND OUTPUT MOLECULES

In Figure 5, we plot the distribution of similarities between input molecules $x_{\text{in}}$ and retrieval $x_R$, intermediate $x_1$, and output molecules $x_{\text{out}}$ using ChatDrug-Turbo. Here we provide more results. The distributions of 8 single-objective small molecule editing tasks can be found in Figure 9, and 6 multi-objective small molecule editing tasks can be found in Figure 10.

As shown in Figures 9 and 10, the observation of similarity distribution satisfies for all 8 single-objective and 6 multi-objective tasks.

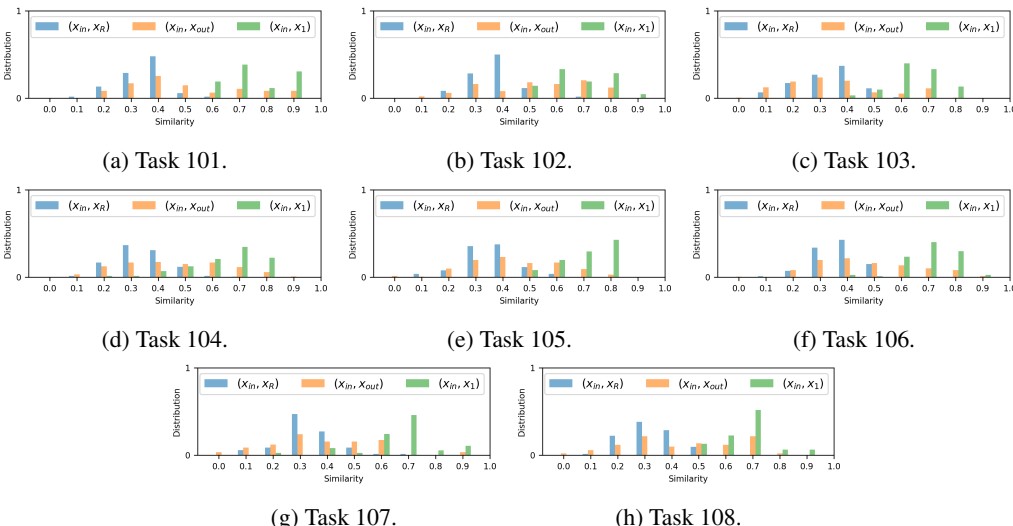

Figure 9: Similarity distribution between input molecules $x_{\text{in}}$ and retrieval $x_R$, intermediate $x_1$, and output molecules $x_{\text{out}}$. Here we show the distribution of 8 single-objective editing tasks.

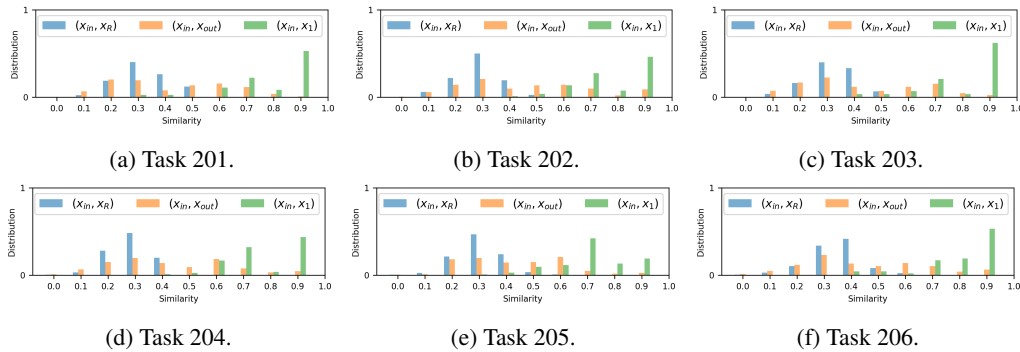

Figure 10: Similarity distribution between input molecules $x_{\text{in}}$ and retrieval $x_R$, intermediate $x_1$, and output molecules $x_{\text{out}}$. Here we show the distribution of 6 multi-objective editing tasks.

## H.2  PEPTIDE

In the main body, we have illustrated how the motif of peptides changes for two peptide editing tasks. Here we show all 6 single-objective editing tasks in Figures 11 to 16.

- For task 301 in Figure 11, ChatDrug-Turbo can successfully upweight E (Glutamic acid) at position 2.
- For task 302 in Figure 12, ChatDrug-Turbo can successfully upweight A (Alanine) at position 2, and L (Leucine) at position 9.
- For task 303 in Figure 13, ChatDrug-Turbo can successfully upweight E (Glutamic acid) at position 2, and L (Leucine) at position 9.
- For task 304 in Figure 14, ChatDrug-Turbo can successfully upweight R (Arginine) and K (Lysine) at position 5, and L (Leucine) at position 9.
- For task 305 in Figure 15, ChatDrug-Turbo can successfully upweight R (Arginine) and K (Lysine) at position 5, and L (Leucine) at position 9.
- For task 306 in Figure 16, ChatDrug-Turbo can successfully upweight E (Glutamic acid) at position 2, and L (Leucine) at position 9.

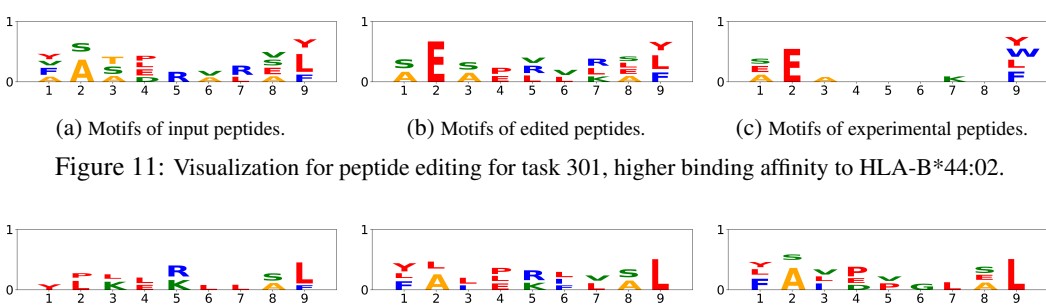

(a) Motifs of input peptides.     (b) Motifs of edited peptides.     (c) Motifs of experimental peptides.

Figure 11: Visualization for peptide editing for task 301, higher binding affinity to HLA-B*44:02.

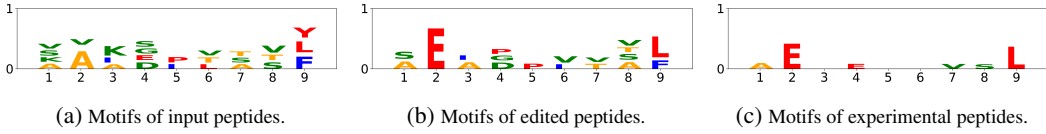

(a) Motifs of input peptides.     (b) Motifs of edited peptides.     (c) Motifs of experimental peptides.

Figure 12: Visualization for peptide editing for task 302, higher binding affinity to HLA-C*03:03.

(a) Motifs of input peptides.     (b) Motifs of edited peptides.     (c) Motifs of experimental peptides.

Figure 13: Visualization for peptide editing for task 303, higher binding affinity to HLA-B*40:01.

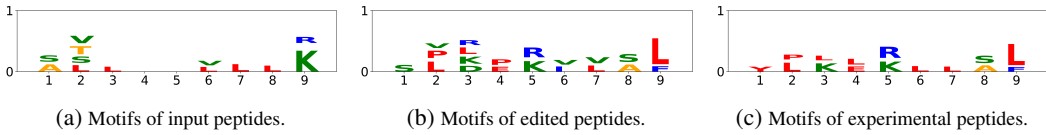

(a) Motifs of input peptides.     (b) Motifs of edited peptides.     (c) Motifs of experimental peptides.

Figure 14: Visualization for peptide editing for task 304, higher binding affinity to HLA-B*08:01.

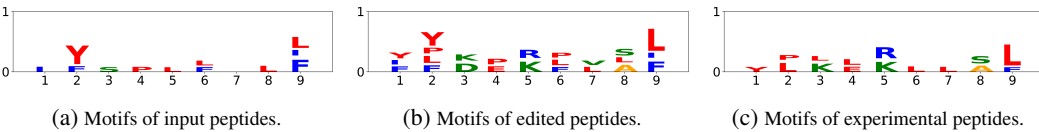

(a) Motifs of input peptides.     (b) Motifs of edited peptides.     (c) Motifs of experimental peptides.

Figure 15: Visualization for peptide editing for task 305, higher binding affinity to HLA-B*08:01.

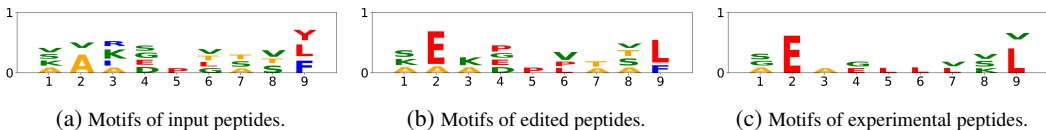

(a) Motifs of input peptides.     (b) Motifs of edited peptides.     (c) Motifs of experimental peptides.

Figure 16: Visualization for peptide editing for task 306, higher binding affinity to HLA-B*40:02.

Here we show all 3 multi-objective editing tasks in Figures 17 to 19. Notice that here there are two target allele types, and we mark them as "target allele 1" and "target allele 2".

- For task 401 in Figure 17, ChatDrug-Turbo can successfully upweight R (Arginine) and K (Lysine) at position 5, and L (Leucine) and F (Phenylalanine) at position 9 for target allele type 1. ChatDrug can also upweight L (Leucine) at position 7, and V (Valine) and L (Leucine) at position 9 for target allele type 2.
- For task 402 in Figure 18, ChatDrug-Turbo can successfully upweight E (Glutamic acid) at position 2, and L (Leucine) at position 9 for target allele type 1. ChatDrug can also upweight F (Phenylalanine) and L (Leucine) at position 9 for target allele type 2.
- For task 403 in Figure 19, ChatDrug-Turbo can successfully upweight R (Arginine) and K (Lysine) at position 5, and L (Leucine) at position 9 for target allele type 1.

**Potential issue on conflicts among target allele types.** One potential challenge is that for multi-objective editing, the target allele types could have conflicting motifs, like the two target alleles for task 403. We leave this for future exploration.

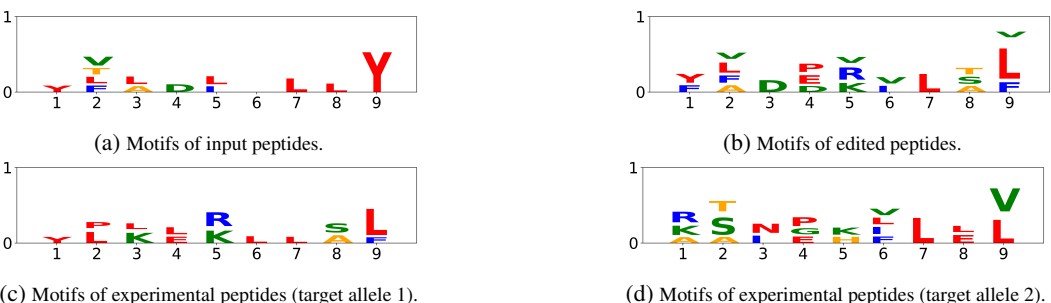

(a) Motifs of input peptides.

(b) Motifs of edited peptides.

(c) Motifs of experimental peptides (target allele 1).

(d) Motifs of experimental peptides (target allele 2).

Figure 17: Visualization for peptide editing for task 401, higher binding affinity to HLA-B*08:01 and HLA-C*15:02.

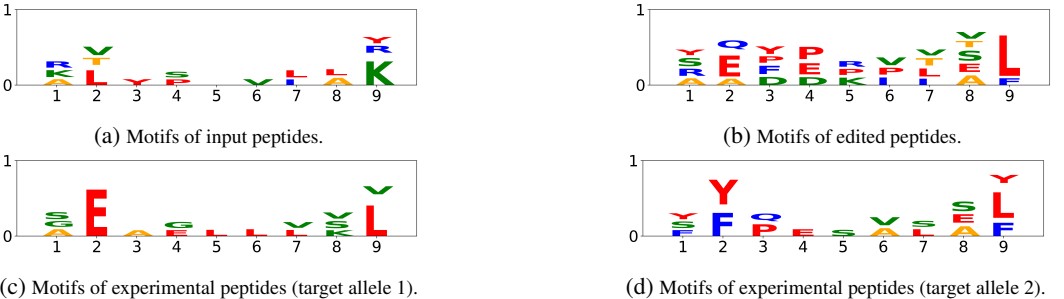

(a) Motifs of input peptides.

(b) Motifs of edited peptides.

(c) Motifs of experimental peptides (target allele 1).

(d) Motifs of experimental peptides (target allele 2).

Figure 18: Visualization for peptide editing for task 402, higher binding affinity to HLA-B*40:02 and HLA-C*14:02.

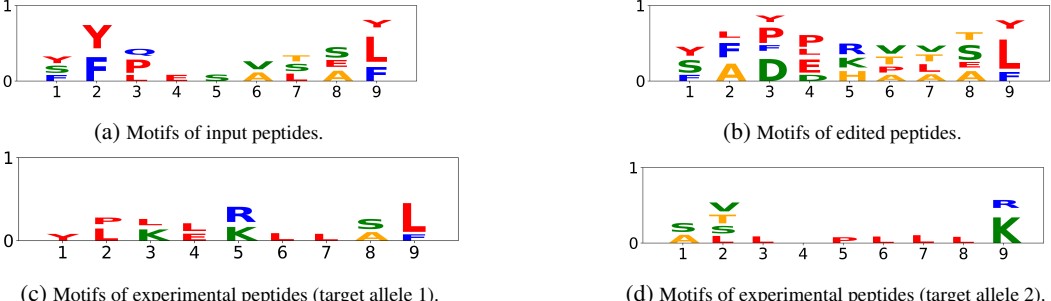

(a) Motifs of input peptides.

(b) Motifs of edited peptides.

(c) Motifs of experimental peptides (target allele 1).

(d) Motifs of experimental peptides (target allele 2).

Figure 19: Visualization for peptide editing for task 403, higher binding affinity to HLA-B*08:01 and HLA-A*11:01.

## H.3 PROTEIN

Recall that we consider two types of secondary structures for protein editing tasks. Both the inputs and outputs are protein sequences. Then we use ESMFold (Lin et al., 2022) for protein folding (protein sequence to protein structure prediction) and then plot the protein structures using PyMOL (Schrödinger & DeLano). For all the protein structure visualizations, we mark $\alpha$-helix structures and $\beta$-strand structures. The edited regions are highlighted in the blue circles.

**Task 501: edit proteins with more helix structures.**

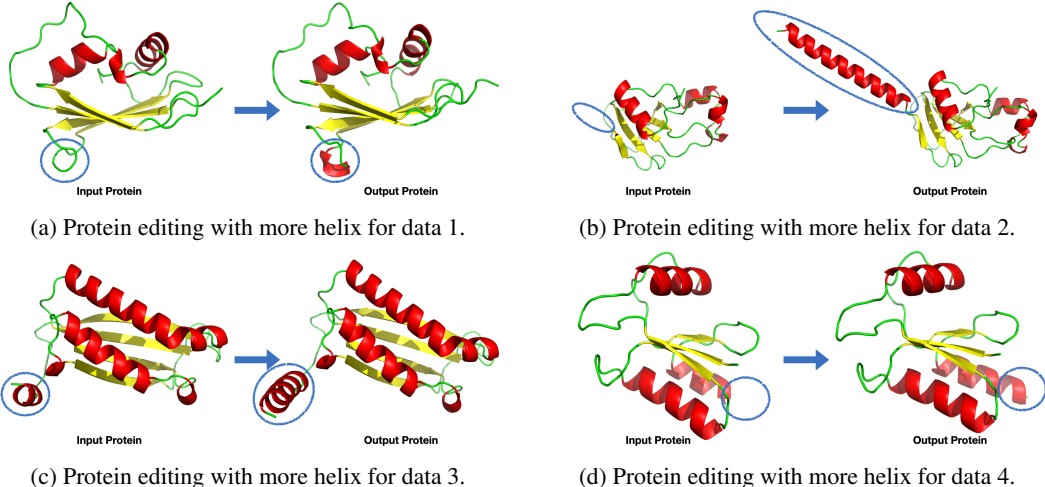

(a) Protein editing with more helix for data 1.   (b) Protein editing with more helix for data 2.

(c) Protein editing with more helix for data 3.   (d) Protein editing with more helix for data 4.

Figure 20: Protein editing with more $\alpha$-helix structures.

**Task 502: edit proteins with more strand structures.**

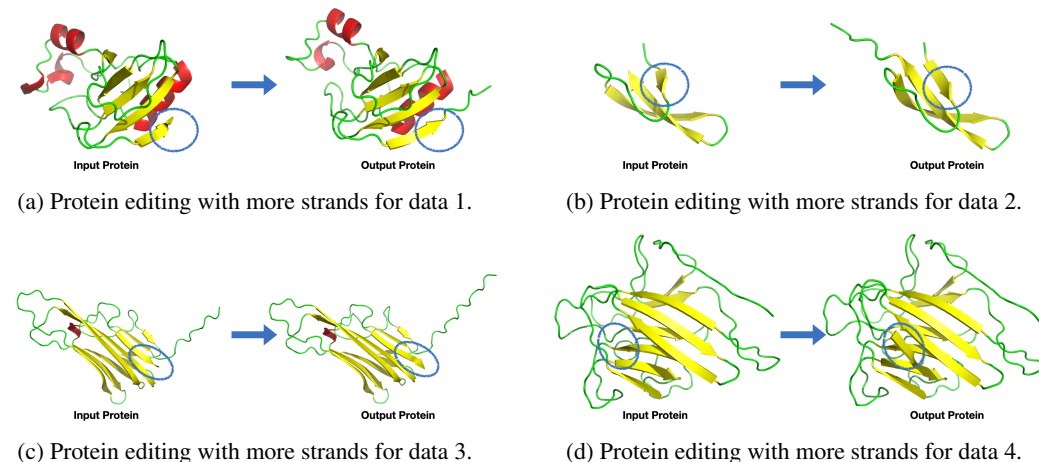

(a) Protein editing with more strands for data 1.   (b) Protein editing with more strands for data 2.

(c) Protein editing with more strands for data 3.   (d) Protein editing with more strands for data 4.

Figure 21: Protein editing with more $\beta$-strand structures.

# I  ABLATION STUDIES

## I.1  ZERO-SHOT AND IN-CONTEXT LEARNING FOR PROTEIN AND PEPTIDE

In Section 4.5, we conduct an ablation study on small molecules to show the comparison between the zero-shot, in-context learning, and ChatDrug-Turbo. Here we conduct the same ablation study on peptides and proteins as follows.

Table 21: Results on six single-objective and three multi-objective peptide editing tasks. Random Mutation-$R$ for $R$ mutated positions. The evaluation is the hit ratio of the increased binding affinity score. The best results are marked in **bold**. Due to the space limitation, please check Appendix F for the text prompt of each task.

| | single-objective editing | | | | | | multi-objective editing | | |
|---|---|---|---|---|---|---|---|---|---|
| | 301 | 302 | 303 | 304 | 305 | 306 | 401 | 402 | 403 |
| Random Mutation-1 | 1.80 | 14.40 | 1.80 | 1.80 | 12.00 | 5.60 | 3.20 | 0.80 | 0.40 |
| Random Mutation-2 | 1.80 | 13.40 | 2.80 | 3.00 | 8.40 | 4.40 | 2.20 | 0.60 | 1.20 |
| Random Mutation-3 | 1.80 | 9.40 | 2.40 | 4.20 | 9.00 | 3.80 | 3.00 | 0.60 | 0.80 |
| In-context Learning (few-shot) | 24.05 | 38.40 | 27.40 | 32.00 | 45.50 | 32.80 | 29.20 | 17.47 | 14.40 |
| ChatDrug ($C = 0$, zero-shot) | 1.60 | 16.80 | 2.40 | 8.22 | 15.00 | 8.02 | 5.41 | 2.00 | 1.20 |
| ChatDrug ($C = 2$) | **56.60** | **69.80** | **64.33** | **59.04** | **65.00** | **64.13** | **44.69** | **34.54** | **41.77** |

Table 22: Results on two protein editing tasks. Random Mutation-$R$ for $R$ mutated positions. The evaluation is the hit ratio of increased secondary structures accordingly. The best results are marked in **bold**.

| | 501 more helix | 502 more strand |
|---|---|---|
| Random Mutation-1 | 18.32 | 17.35 |
| Random Mutation-2 | 24.95 | 19.69 |
| Random Mutation-3 | 26.90 | 21.44 |
| In-context Learning (few-shot) | **36.64** | 44.47 |
| ChatDrug ($C = 0$, zero-shot) | 21.43 | 23.73 |
| ChatDrug ($C = 2$) | 33.18 | **59.68** |

## I.2 ABLATION STUDY ON THE THRESHOLDS IN FEEDBACK CONDITION FUNCTION

In the main body, we conduct an ablation study on the thresholds in the feedback condition function. Due to the space limitation, we only list the mean in Table 7. Here we list both the mean and standard deviation as follows.

Table 23: Ablation studies on single-objective small molecule editing and feedback condition $D$ with five seeds and two conversational rounds. The evaluation metric $E$ uses the strict threshold for each task.

|  | 101 | 102 | 103 | 104 | 105 | 106 | 107 | 108 |
|---|---|---|---|---|---|---|---|---|
| loose threshold | 80.73±1.32 | 41.00±0.91 | 11.23±2.70 | 16.94±1.24 | 33.16±2.22 | 53.59±1.59 | 14.96±1.96 | 21.93±1.82 |
| strict threshold | 88.67±0.95 | 70.08±3.44 | 19.37±5.54 | 30.99±2.66 | 43.08±2.95 | 66.69±2.74 | 72.60±2.51 | 76.43±3.32 |

Table 24: Ablation studies on multi-objective small molecule editing and feedback condition $D$ with five seeds and two conversational rounds. The evaluation metric $E$ uses the strict threshold for each task.

|  | 201 | 202 | 203 | 204 | 205 | 206 |
|---|---|---|---|---|---|---|
| loose threshold | 20.14±0.86 | 7.96±2.05 | 17.93±0.79 | 5.79±1.38 | 3.66±0.24 | 41.04±1.66 |
| strict threshold | 49.64±2.66 | 24.92±4.85 | 53.64±5.81 | 24.19±2.19 | 10.44±5.75 | 52.9±2.23 |

### I.3 ABLATION STUDY ON THE NUMBER OF REQUEST ANSWERS IN ZERO-SHOT CHATDRUG-TURBO

Notice that in Table 9, we list five molecules (a.k.a. five trials) for each answer. In this subsection, we would like to conduct an ablation study to explore in the zero-shot setting of ChatDrug-Turbo, *i.e.*, with the conversation round $C = 0$, if we can obtain higher performance using more trial numbers. This means that for each input small molecule, we have five edited small molecules; as long as one of them is a hit, then we say this is a successful hit. The results for 14 tasks with the loose threshold are listed below.

Table 25: Ablation studies on different trial numbers on single-objective molecule editing, with $C = 0$ and seed is 0.

|  | loose condition $\Delta = 0$ | | | strict condition $\Delta > 0$ | | |
|---|---|---|---|---|---|---|
|  | trial = 1 | trial = 3 | trial = 5 | trial = 1 | trial = 3 | trial = 5 |
| 101 *more soluble in water* | 78.26 | 88.77 | 93.05 | 68.48 | 80.21 | 85.03 |
| 102 *less soluble in water* | 71.35 | 89.95 | 93.12 | 24.16 | 74.60 | 78.84 |
| 103 *more like a drug* | 16.15 | 45.64 | 53.81 | 2.08 | 4.62 | 7.11 |
| 104 *less like a drug* | 32.12 | 68.37 | 75.00 | 2.07 | 17.86 | 31.12 |
| 105 *higher permeability* | 16.04 | 27.98 | 33.16 | 9.63 | 18.13 | 22.28 |
| 106 *lower permeability* | 8.33 | 34.04 | 57.67 | 5.56 | 24.47 | 42.86 |
| 107 *more hydrogen bond acceptors* | 59.41 | 76.57 | 83.15 | 1.76 | 18.29 | 33.71 |
| 108 *more hydrogen bond donors* | 63.16 | 85.23 | 89.77 | 5.85 | 19.89 | 32.39 |

Table 26: Ablation studies on different trial numbers on multi-objective molecule editing, with $C = 0$ and seed is 0.

|  | loose condition $\Delta = 0$ | | | strict condition $\Delta > 0$ | | |
|---|---|---|---|---|---|---|
|  | trial = 1 | trial = 3 | trial = 5 | trial = 1 | trial = 3 | trial = 5 |
| 201 *more soluble in water* and *more hydrogen bond acceptors* | 43.09 | 69.89 | 75.40 | 6.08 | 22.04 | 34.22 |
| 202 *less soluble in water* and *more hydrogen bond acceptors* | 0.52 | 13.47 | 31.44 | 0.00 | 0.52 | 2.06 |
| 203 *more soluble in water* and *more hydrogen bond donors* | 54.49 | 79.67 | 81.97 | 6.18 | 18.13 | 35.52 |
| 204 *less insoluble in water* and *more hydrogen bond donors* | 0.53 | 5.21 | 13.54 | 0.00 | 0.00 | 1.04 |
| 205 *more soluble in water* and *higher permeability* | 2.11 | 4.15 | 5.18 | 1.05 | 1.04 | 1.04 |
| 206 *more soluble in water* and *lower permeability* | 22.22 | 59.07 | 69.43 | 16.40 | 48.70 | 55.44 |

## I.4 ABLATION STUDY ON REASONING AND KNOWLEDGE EXTRACTION OF CHATDRUG

In Figure 4, we show the knowledge extraction ability of ChatDrug. As a control experiment, here we test the knowledge extraction ability of ChatDrug without the examples illustrated in the conversation. As shown in Figure 22, the extracted knowledge is more general and less informative when we remove the examples.

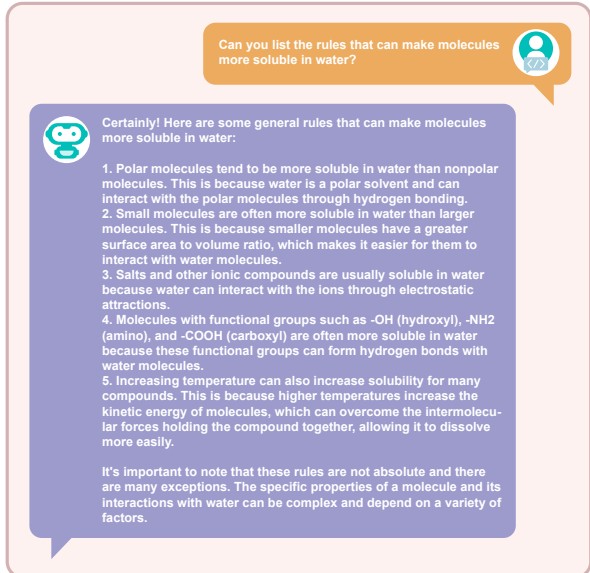

Figure 22: Knowledge extraction directly without any conversations.

Then as discussed previously, the extracted information in ChatDrug contains redundant information. For example, the one listed in Figure 22 extracts three rules with overlap meanings. Then we conduct a further ablation study by forcing ChatDrug to extract three rules from the five original rules in Figure 4. We observe that ChatDrug successfully merges rule 1 and rule 5, both of which express the same concept that polar functional groups lead to good solubility. Interestingly, ChatDrug also merges rule 3 with rule 4. Though these 2 rules share the idea of ring structure, the essence of rule 3 is introducing polar substituents. Thus, rule 3 should be more relevant to rules 1 and 5. Nevertheless, it is clear that ChatDrug understands the domain knowledge and is capable of extracting and summarizing it. Overall, we would like to give positive feedback on ChatDrug.

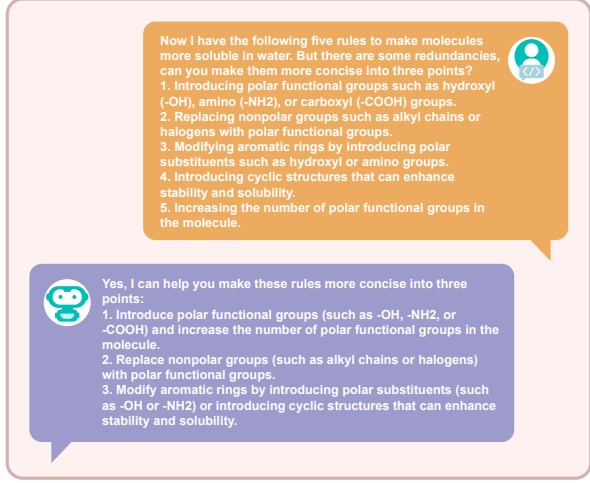

Figure 23: Further knowledge extraction.

To sum up, we observe that ChatDrug can understand and extract the knowledge to some extent, though not perfectly. We believe this is a promising direction for future exploration.

