# OpenReview forum: "Conversational Drug Editing Using Retrieval and Domain Feedback"
_ICLR.cc/2024/Conference — ICLR 2024 poster_

### Official Review · Reviewer_WoGj · 2023-10-29

**Soundness:** 3 good
**Presentation:** 2 fair
**Contribution:** 2 fair
**Rating:** 6
**Confidence:** 2

**Summary:**

This paper proposes a framework named ChatDrug that leverages large language models to streamline effective drug editing. The authors demonstrate that ChatDrug outperforms existing methods on all 39 drug editing tasks and can offer insightful explanations from a domain-specific perspective. The authors comprehensively evaluate of ChatDrug's performance on 39 drug editing tasks, and conduct a detailed analysis of ChatDrug's ability to provide domain-specific explanations for its decisions.

**Strengths:**

1.  This is the first paper to leverages large language models for effective drug editing.
2. The authors conduct comprehensive evaluation of ChatDrug's performance on drug editing tasks, as well as the analysis of explainability.

**Weaknesses:**

1. The domain feedback function in the ReDF is defined as the evaluation metric. I wonder if it may potentially cause 'information leakage', as it is accessing the information of 'success sequence editing'.
2. While the detailed implementation of the modules are new, the paradigm is lack novelty, as it falls into the 'prompt, retrieval for factuality, evaluate and repeat' paradigm, which is not new.
3. The paper includes much domain knowledge for demonstration, which causes troublesome in comprehending the main idea. The authors may consider simplify the terms and focus on the main experimental phenomenon only.

**Questions:**

Refer to weaknesses.

---

> ### Author Response · Authors · 2023-11-21
>
> Thank you for acknowledging our work as the first to apply LLM for effective drug editing with a comprehensive evaluation. We hope that we have adequately addressed your concerns, and we are happy to provide more explanations.
>
> **Q: Concerns about information leakage**
>
> Thank you for raising this question. First, we would like to claim that from the data aspect, there is no information leakage. The question is indeed about the evaluation, and we agree that there could exist evaluation bias. However, coming up with an accurate and computationally feasible evaluation metric is still an open challenge in the AI for science research field. If the community has developed any better evaluation metrics, they can be incorporated into the ChatDrug pipeline smoothly. Besides in our work, to alleviate this issue, we have tried our best to include the qualitative analysis on all 39 tasks.
>
> **Q: Novelty about the prompt and retrieval paradigm**
>
> On one hand, we want to claim that combining LLM with prompt, retrieval, and conversation has attracted attention recently. Specifically, the methods of *prompt engineering and information retrieval have been widely applied to enhance LLMs* (Wei et al. 2022; Izacard et al. 2022). Meanwhile, the advent of state-of-the-art, *instruction-following LLMs this year* (e.g. ChatGPT 2022 and LLaMA2-Chat 2023) marks a significant advancement in the conversational abilities of these models. The *integration of LLMs' conversational ability with prompt and retrieval techniques* represents a novel and emerging paradigm. This innovative approach has been adopted to further improve the performance of LLMs in a variety of NLP tasks in the past months (Madaan et al. 2023; Peng et al. 2023).
>
> On the other hand, our work is the first to prove that such a novel paradigm *(LLMs w/ prompt, retrieval, and conversation)* can indeed work in AI for drug design problems. This is inspiring for the whole community, and has the potential to lead the next wave of research interesting on using LLMs for science exploration.
>
> *Wei, Jason, Xuezhi Wang, Dale Schuurmans, Maarten Bosma, Fei Xia, Ed Chi, Quoc V. Le, and Denny Zhou. Chain-of-thought prompting elicits reasoning in large language models. Advances in Neural Information Processing Systems 35 (2022): 24824-24837.*
>
> *Izacard, Gautier, Patrick Lewis, Maria Lomeli, Lucas Hosseini, Fabio Petroni, Timo Schick, Jane Dwivedi-Yu, Armand Joulin, Sebastian Riedel, and Edouard Grave. Few-shot learning with retrieval augmented language models. arXiv preprint arXiv:2208.03299 (2022).*
>
> *Madaan, Aman, Niket Tandon, Prakhar Gupta, Skyler Hallinan, Luyu Gao, Sarah Wiegreffe, Uri Alon et al. Self-refine: Iterative refinement with self-feedback. arXiv preprint arXiv:2303.17651 (2023).*
>
> *Peng, Baolin, Michel Galley, Pengcheng He, Hao Cheng, Yujia Xie, Yu Hu, Qiuyuan Huang et al. Check your facts and try again: Improving large language models with external knowledge and automated feedback. arXiv preprint arXiv:2302.12813 (2023).*
>
> **Q: Simplify the terminologies and interpretation**
>
> We apologize for any inconvenience our work may have caused you in your reading experience. When we wrote the paper, we tried to add visual explanations and qualitative analysis for all the terminologies and results, for example, Figure 1, Table 3, and Figures 2-4. We are aware that reaching a good balance between the simplicity-and-expertness trade-off is important, especially if we want to show the potential of our work to the ML community. Thus, if you can help list the specific sentences that confuse you, we are happy to answer them and improve the understandability of our work.

---

> > ### Comment · Reviewer_WoGj · 2023-11-22
> > **Response to the Authors**
> >
> > Thank you for your response. I have carefully read them and acknowledge the contributions. However, I have remained concerns about the novelty ChatDrug. Applying an existing SOTA LLM paradigm (when it is novel) to a new domain while addressing the domain-specific questions is, from my perspective, not enough. I believe this has great potential to the industry, but marginal to the research community. Therefore, I would like to remain my score. Thanks.

---

> > > ### Author Response · Authors · 2023-11-23
> > > **Official Comment by Authors**
> > >
> > > Hi reviewer WoGj,
> > >
> > > Thank you for sharing your thoughts, and we completely respect your comments. On the other hand, we would like to share more insights from our side.
> > >
> > > 1. Applying existing methods for solving a new task for the first time is indeed novel and important as well (as detailed in this [blog](https://medium.com/@black_51980/novelty-in-science-8f1fd1a0a143)), especially in the AI for drug discovery community. For example:
> > >     - **molecule representation**: Graphormer = invariant + Transformer [1]
> > >     - **protein pretraining**: TAPE = protein sequence + Masked Language Model [2]
> > >     - **conformation generation**: EDM = DDPM + EGNN [3]
> > > If you into the details of these works, the main contribution is not the technical parts, since the key modules (Transformer, MLM, DDPM, and EGNN) are well-studied. Instead, the main contribution is to open a new research direction and show how to adapt existing tools to make them work on new problems. In other words, the problem definition, the pipeline, and engineering works are the main contributions in this case.
> > >
> > > 2. From the domain perspective, first thank you for acknowledging that our problem is important. Then we want to point out that, our work is the first to show that using this novel  LLM paradigm (prompt-retrieval-domain-conversation) can exhibit promising results in drug editing, which is one of the most important tasks in both academia and industry. Such domain-specific insights are inspiring for all the researchers in drug discovery, especially in bringing more attention to AI for drug discovery direction; following researchers can explore if such an LLM paradigm can work for other challenging tasks, like protein folding, pathway detection, docking, etc.
> > >
> > > 3. From the technical details, we also want to claim that ChatDrug pipeline does not simply take the existing paradigm; instead, ChatDrug possesses certain domain-specific novelties, as listed below:
> > >     - The retrieval function is specifically designed to take both the domain feedback and similarity function into consideration.
> > >     - The ablation studies of in-context learning reveal its effectiveness.
> > >     - We are aware that one bottleneck in the AI for drug discovery research line is the evaluation. Thus, we carefully choose three types of tasks on small molecules, proteins, and peptides, with computationally feasible evaluation metrics.
> > >     - Additionally, we also qualitatively provide the visual analysis of these tasks. Before ChatDrug, the community hadn’t found a good way of showing the ability of LLM for drug discovery, and we achieved this both quantitatively and qualitatively.
> > >
> > > We hope this provides further insights into our work, and we would greatly appreciate any reconsideration of the assigned score.
> > >
> > > ---------
> > >
> > > [1] Do Transformers Really Perform Bad for Graph Representation?, NeurIPS 2021
> > >
> > > [2] Evaluating Protein Transfer Learning with TAPE, NeurIPS 2019
> > >
> > > [3] Equivariant Diffusion for Molecule Generation in 3D, ICML 2022

---

> > > > ### Comment · Area_Chair_VNGE · 2023-12-02
> > > > **Does the response address your concern about novelty?**
> > > >
> > > > @Reviewer WoGj,
> > > >
> > > > I would appreciate it if you could review the response and adjust your comments (and rating) as necessary.
> > > >
> > > > AC

---

> > > > ### Comment · Reviewer_WoGj · 2023-12-04
> > > > **Response to the Authors**
> > > >
> > > > Thanks for the further clarification. I have carefully reviewed your response and adjusted my scores.

---

### Official Review · Reviewer_XYSR · 2023-10-31

**Soundness:** 3 good
**Presentation:** 3 good
**Contribution:** 3 good
**Rating:** 6
**Confidence:** 3

**Summary:**

The paper proposes ChatDrug, a framework for conversational drug editing using large language models (LLMs). ChatDrug leverages prompt design, retrieval and domain feedback, and conversation modules to generate diverse and valid suggestions for drug editing. ChatDrug can handle various types of drugs, such as small molecules, peptides, and proteins. The paper evaluates ChatDrug on 39 drug editing tasks and shows that it outperforms several baselines.

**Strengths:**

1. The paper addresses an important and challenging problem of drug editing using LLMs.
2. The paper introduces a novel and comprehensive framework that incorporates domain knowledge and interactive feedback for drug editing.

**Weaknesses:**

1. The paper does not provide a clear comparison or analysis of the different LLM backbones used in ChatDrug.
2. The paper does not provide any user study or evaluation from domain experts to validate the usefulness and usability of ChatDrug.

**Questions:**

1. How do you ensure the quality and reliability of the retrieval and domain feedback module? How do you handle the cases where the retrieved information is inaccurate or outdated?
2. How do you measure the similarity between the input and output drugs? How do you balance the trade-off between similarity and diversity in drug editing?

---

> ### Author Response · Authors · 2023-11-21
>
> Thank you for acknowledging our work as important, novel, and with comprehensive results. We hope that we have addressed your concerns, and any re-evaluation of our work is appreciated.
>
> **Q: Comparison and analysis of different LLM backbones**
>
> Thank you for raising this question. For the main results, we have comparison results of three different LLMs as backbone: Galactica, Llama2, Turbo (ChatGPT) in the first version. Through the comparison, we find that the Turbo (ChatGPT) backbone can help ChatDrug achieve the best performance and highest stability for all three drug types. ChatDrug with GALACTICA as backbone only performs better than baseline on the small molecule editing and peptide editing but cannot deal with protein editing tasks. ChatDrug with Llama2 as backbone can achieve better performance than baseline methods on peptide editing and protein editing but fails to cover small molecule editing tasks. Such unstable performance across different drug types may be attributed to the limited domain specific training data for open-source LLMs like GALACTICA and Llama2. You can find the details in Table 1, 2, 4, 5.
>
>
>
>
> **Q: Validation from domain experts.**
>
> Thank you for asking this question. We completely agree that for works like ChatDrug, validation from the domain experts is definitely important. Thus, in our manuscript, we have done the following analysis:
> - For small molecules (Table 3), we show how the molecule substructures change using ChatDrug. For example, in Table 3a, through conversational guidance, ChatDrug-Turbo changes the methyl group of the input molecule to an aminoethyl group, successfully obtaining a less soluble molecule.
> - For peptides (Figure 2), we show how the motifs change using ChatDrug.  For instance, for task 301 (editing input peptide binding to HLA-B*44:02), the edited peptides can successfully upweight the alphabet E (glutamic acid) at position 2.
> - For proteins (Figure 3), we show how the secondary structures change using ChatDrug. The optimized parts (alpha-helix and beta-sheet) are highlighted in blue circles.
>
> **Q: Handling the inaccurate or outdated retrieved information**
>
> Thank you for raising such an interesting question! We had the same concern when we wrote the paper, and that’s why we qualitatively show three molecules for each editing task in Table 3.
>
> - As shown in Eq 2, the retrieved information is guaranteed to be accurate. However, this does not mean they always have a positive effect.
> - In another way, your question can be translated to “Can ChatDrug handle cases where the retrieved information has a negative impact?” The answer is yes. In Table 3, for each editing task, we show three molecules: input (blue), intermediate and wrong molecules (red), and the final output molecules (green). The intermediate molecules do not satisfy our conditions, yet ChatDrug possesses the ability to correct this in an iterative manner.
>
> Thank you for the in-depth discussion again.
>
> **Q: Concerns about trade-off between similarity and diversity**
>
> This is another critical question. So for now, we mainly balance the similarity and diversity through prompt engineering. For example in the LLM, we added a phrase in the prompt, requesting “the output drugs should be similar to the input drugs” (Appendix F). For the ReDF evaluation function (Eq 2), we use Tanimoto similarity for small molecules and edit distance for peptides and proteins.

---

### Official Review · Reviewer_pPSW · 2023-11-03

**Soundness:** 3 good
**Presentation:** 3 good
**Contribution:** 3 good
**Rating:** 6
**Confidence:** 4

**Summary:**

The paper presents an approach to editing small molecules / peptides / proteins by interacting with an out-of-the-box LLM such as ChatGPT Turbo.

They describe how to prompt the model and then how to give feedback using a small number of supervised examples  using a retrieval and domain feedback module which finds the positive example that is closest to the negative prediction.

The results are strong across 3 tasks that involve small molecules, peptides and proteins but I have some questions / concerns.

**Strengths:**

-The paper presents an interesting approach to drug editing that leverages pretrained LLMs. It is interesting to know that non-protein LLMs out of the box can reason about small molecules / proteins / peptides through iterative in-context learning.

**Weaknesses:**

After the author response, the authors answered my concern (1).

For (2), I emphasized that I wasn't suggesting comparing with supervised baselines on a supervised task, but rather reframing some of the datasets as a few shot task could increase the impact of the work (but I agree may be a significant challenge since it involves quantitative measurements).

Accordingly I have now increased my score.

-----

(1) I am not sure about the small molecules experiment but atleast for the other two, the only baselines provided are random. However, I don't feel this is fair since the author's approach is seeing a few positive examples. Would be great to have a baseline that uses a similar number of examples.

(2) The tasks focus on some basic properties of molecules like water solubility. However, often what we are really interested in drug design is a quantitative measurement like the binding affinity to a given target (or something similar). I don't see any results along these lines.

In small molecules there exist benchmarks that measure the binding to specific targets or other more detailed attributes. For example the datasets/baselines used in this paper:

https://arxiv.org/pdf/2206.07632.pdf

----

(Maybe less related since the paper's focus seems to be more on small molecules with proteins a secondary experiment) For  proteins there exist benchmarks like Deep Mutational Scanning and FLIP and associated works that try to optimize a protein towards one of these attributes.

https://www.biorxiv.org/content/10.1101/2021.11.09.467890v1

https://www.biorxiv.org/content/10.1101/2021.07.09.450648v2

https://arxiv.org/abs/2303.04562

https://arxiv.org/abs/2307.00494

**Questions:**

I am curious as to what baselines the authors think are fair for each task that use similar amounts of supervised data. This is a bit unclear to me in the paper.

---

> ### Author Response · Authors · 2023-11-21
>
> Thank you for recognizing the interesting pipeline and strong results of our work. Your concerns are critical and indeed related to our big picture. We hope our explanations have adequately addressed your questions.
>
> **Q: More baselines for peptides and proteins.**
>
> First thank you for raising this question. Our contents are a little dense, so we move some of our discussions into the appendix in the first version. We are very happy to explain more details to you, and hope the following answers can solve your questions.
> - In Appendix I, we do provide the zero-shot ($C=0$) editing results on peptides and proteins.
> Editing performance of the zero-shot setting is as bad as the random mutation baseline. For example, the hit ratio of task 502 (more strand for protein editing) for zero-shot (23.73%) is only slightly higher than the random mutation (21.44%).
> - However, one advantage of our ChatDrug pipeline is that we can keep improving the baselines in a few-shot manner, by utilizing the retrieval and conversation module. We can further improve the hit ratio to 59.68% through the ChatDrug pipeline for task 502. This is one of the main advantages of ChatDrug.
> - On the other hand, the methods using positive and negative samples are called *supervised controllable generation* (you also mentioned relevant papers in the following, thanks again). But this is slightly different from the few-shot setting ($C$-shot) in ChatDrug, because it is hard to train a powerful classifier on less than 5 positive samples. Thus, we want to claim that such a few-shot editing is indeed one advantage of ChatDrug.
> - We want to highlight that in ChatDrug, the main goal is to prove that such a novel paradigm (in-context and conversation) can lead to more accurate performance on drug design tasks.
>
> **Q: Experiments of binding tasks.**
>
> The problem is very important but challenging. The first challenge is about how to get an accurate textual description, since it may relate to the 3D structures of ligands and proteins. The next challenge is the dataset. For instance, if we take the protein-ligand (small molecule) binding from PDB, there are only a few ligands that can bind to the target proteins, i.e., we cannot construct a sufficiently large retrieval database for the few-shot setting in ChatDrug.
>
> **Q: Related papers about molecular and protein supervised editing.**
>
> Thank you for raising this concern, this is interesting for future exploration and is slightly different from our current focus. We will explain this in detail below.
> -  The MOOD paper proposes the score-matching method with OOD control, where the control is a classifier pretrained on the whole ZINC dataset. In other words, this is the supervised editing task, which needs a classifier-guided generation. The key to success is how to train a powerful classifier.
> - This also holds for the protein editing/optimization papers shared by you.
> - However, in ChatDrug, we retrieve only $C$ small molecules/proteins after $C$ rounds of conversation. Our setting is of the few-shot ($C$-shot) editing.
> - Last but not least, although these papers are in different settings, supervised editing (like MOOD) is definitely interesting to incorporate into ChatDrug. We added this in our revised manuscript (Conclusion section).
>
> **Q: Fair baselines in LLM setting.**
>
> It is challenging for us to add baselines in the first place, because using LLM for few-shot editing is a very novel paradigm, and we are actually creating a new research line. Thus, we would claim that the most fair comparison would be to use ChatDrug with different LLMs for the same task setting. This is also the reason why we introduced three types of backbone LLMs (Galactica, Llama2, and Turbo/ChatGPT) in the ChatDrug pipeline.

---

> ### Comment · Reviewer_pPSW · 2023-11-22
> **Thank you for your response**
>
> Hi,
>
> Thanks for your reply. Just as a clarification, when I suggested the related works I wasn't suggesting that you directly compare with them since they are supervised. I was suggesting converting some of those tasks into few shot tasks (perhaps using the training set as the database to retrieve from).
>
> However, I do appreciate the response and thus have increased my score.

---

> ### Author Response · Authors · 2023-11-22
> **Thank you for the clarification**
>
> Hi Reviewer pPSW,
>
> Thank you for the clarification! We definitely agree that these tasks can be adapted into our ChatDrug pipeline, using the training set as the retrieval DB. Thank you again for sharing them, and we are happy to include them in our next project.
>
> One thing we may kindly need your help with is boosting the score, as we've noticed it remains unchanged.
>
> Thank you again for the positive feedback!
>
> Regards, Authors of ChatDrug

---

> > ### Comment · Reviewer_pPSW · 2023-11-22
> >
> > Thanks for the note, just updated the review and score.

---

### Meta-Review · Area_Chair_VNGE · 2023-12-11

**Metareview:**

The paper presented an approach for editing small molecules, peptides, and proteins using large language models (LLMs), and received generally positive reviews.

**Justification For Why Not Higher Score:**

This is nice work, however, it does come with certain limitations such as the lack of technical innovation and the need for inclusion of more challenging tasks among others.

**Justification For Why Not Lower Score:**

The work provides solid contributions to the relevant fields.

---

### Decision · Program_Chairs · 2024-01-16

Accept (poster)